# *Pseudolycoriella hygida* (Sauaia and Alves)—An Overview of a Model Organism in Genetics, with New Aspects in Morphology and Systematics

**DOI:** 10.3390/insects15020118

**Published:** 2024-02-06

**Authors:** Frank Menzel, Katja Kramp, Dalton de Souza Amorim, Eduardo Gorab, João Vitor Cardoso Uliana, Heni Sauaia, Nadia Monesi

**Affiliations:** 1Senckenberg Deutsches Entomologisches Institut (SDEI), Eberswalder Straße 90, 15374 Müncheberg, Germany; frank.menzel@senckenberg.de; 2Leibniz Centre for Agricultural Landscape Research (ZALF), Eberswalder Straße 84, 15374 Müncheberg, Germany; katja.kramp@zalf.de; 3Departamento de Biologia, Faculdade de Filosofia, Ciências e Letras de Ribeirão Preto, Universidade de São Paulo, Av. Bandeirantes, 3900, Vila Monte Alegre, Ribeirão Preto 14040-900, SP, Brazil; dsamorim@usp.br; 4Independent Researcher, Rua Marcus Pereira 167/213, São Paulo 05642-020, SP, Brazil; gorabcrow@gmail.com; 5Programa de Biologia Celular e Molecular, Faculdade de Medicina de Ribeirão Preto, Universidade de São Paulo, Av. Bandeirantes, 3900, Vila Monte Alegre, Ribeirão Preto 14049-900, SP, Brazil; jvitoruli@gmail.com; 6Faculdade de Medicina de Ribeirão Preto, Universidade de São Paulo, Av. Bandeirantes, 3900, Vila Monte Alegre, Ribeirão Preto 14049-900, SP, Brazil; 7Departamento de Análises Clínicas, Toxicológicas e Bromatológicas, Faculdade de Ciências Farmacêuticas de Ribeirão Preto, Universidade de São Paulo, Av. do Café, s/n, Vila Monte Alegre, Ribeirão Preto 14040-903, SP, Brazil

**Keywords:** Sciaroidea, Sciaridae, *Pseudolycoriella*, Neotropical Region, cytogenetics, morphology, systematics, literature review

## Abstract

**Simple Summary:**

*Pseudolycoriella hygida* (Sauaia & Alves, 1968) is a lower Diptera species that belongs to the Sciaridae. This family includes about 3000 described species worldwide, including 161 in the genus *Pseudolycoriella*. Sciarids are also commonly known as ‘black fungus gnats’, and in nature their larvae are mainly found in decomposing plant material where fungal activity is apparent. *Pseudolycoriella hygida* is one of the few sciarid species that has been successfully maintained in culture under laboratory conditions for several decades. Since the original culture was established nearly 60 years ago, studies on *Psl. hygida* have contributed to the understanding of different aspects of insect biology. The aim of this work is to deepen our understanding of this species by presenting previously unpublished morphological, cytological, and molecular data. Apart from contributing to the characterization of *Psl. hygida*, this work is an important resource for future studies on the biology of both Sciaridae and Diptera.

**Abstract:**

*Pseudolycoriella hygida* (Sauaia & Alves, 1968) is a sciarid that has been continuously cultured in the laboratory for nearly 60 years. Studies on this species have contributed to the understanding of DNA puffs, which are characteristic of Sciaridae, and to the knowledge of more general aspects of insect biology, including cell death, nucleolar organization, and the role of the hormone ecdysone during molting. The genome of *Psl. hygida* has now been sequenced, and it is the third publicly available sciarid genome. The aim of this work is to expand the current knowledge on *Psl. hygida*. The morphology of the adults is revisited. The morphology of larvae and pupae is described, together with the behavior of immature stages under laboratory conditions. Cytogenetic maps of the salivary gland polytene chromosomes are presented, together with a comparative analysis of the mitotic chromosomes of six different sciarid species. *Pseudolycoriella hygida* was originally described as a species of *Bradysia* and recently moved to *Pseudolycoriella.* We examine here the systematic position of *Psl. hygida* in the latter genus. Our results extend the characterization of an unconventional model organism and constitute an important resource for those working on the cytogenetics, ecology, taxonomy, and phylogenetic systematics of sciarids.

## 1. Introduction

The fly family Sciaridae Billberg, 1820 (black fungus gnats) contains about 3000 described species worldwide [1], of which 161 belong to the genus *Pseudolycoriella* Menzel & Mohrig, 1998 [2,3,4]. Most sciarids are blackish-brown, with grayish or brownish wing membranes, but the bodies of some species are patterned with light brown, orange, or yellowish. The Sciaridae belong to the superfamily Sciaroidea and display a broad geographic distribution, inhabiting diverse environments ranging from sub-arctic to desert regions, with a correspondingly wide range of colonized terrestrial habitats [5,6,7]. There are currently about 300 described species of sciarids in the Neotropical realm, although the classification of many of these species needs to be revised [8]. Further research will certainly increase the number of genera and species recognized in the region, as is the case of *Pseudolycoriella*.

A current hypothesis, based on both morphological and molecular data, is that during the course of evolution, the larvae of some sciarids shifted from feeding on dead plant material (plant litter and/or rotten wood) to feeding on living plants [5]. Abundant decaying vegetable material often leads to high population levels of sciarids, some of which become pests. The larvae of synanthropic species can feed on plants or mushrooms of economic importance, and although increasing attention has been given in the literature to these pest species, they represent only a small fraction of the diversity of this family [9,10].

Sciarids have played a key role in the development of our understanding of genetics; however, to some extent, this contribution has been overlooked in historical accounts of the subject. In many dipteran larval tissues, including the salivary glands, the interphase chromosomes are polytene, which arise from multiple rounds of DNA replication without the separation of the sister chromatids. The quantity of DNA in polytene nuclei increases in geometric steps and is accompanied by an increase in cell size. Polyteny is understood as an effective mechanism that accelerates growth and enhances gene expression [11]. These gigantic chromosomes were initially described by Balbiani [12], and their morphology and chemical nature were elucidated through studies on species of bibionids, *Chironomus* Meigen, 1803, and *Drosophila* Fallén, 1823 in the 1930s (reviewed in [11,13]). Since the 1920s, different sciarid species have been used as model organisms to investigate some key aspects of genetics. An unusual mechanism of chromosome elimination and sex determination in sciarids was initially described in the 1920s by Charles W. Metz (reviewed in [14]). In the 1950s, Marta E. Breuer and Crodowaldo Pavan investigated DNA puffing patterns, the increase in DNA content, and the abundant transcription occurring in DNA puff-forming regions in the salivary gland polytene chromosomes at the end of the fourth larval instar (reviewed in [15,16]). In the following decades, various groups both in Brazil and in the United States worked on the molecular characterization of sciarid DNA puffs, which contributed to the understanding of eukaryotic origins of DNA replication (reviewed in [17,18,19]).

The role of ecdysone in sciarid DNA puff formation, transcription, and amplification was initially investigated by Helen V. Crouse in the 1960s and further expanded upon by other research groups working with various sciarid species as model organisms. Together, these studies contributed to the understanding of the role of ecdysone in DNA puff formation, transcription, and DNA replication in DNA puff-forming regions (reviewed in [20]). Besides the characterization of processes typical of sciarids—namely, the unusual mechanism of sex determination and DNA puff formation—the study of sciarids also contributed to the understanding of telomere organization, nucleolar structure [21], antimicrobial peptides, and cell death [22] (reviewed in [18]). More recently, the research focus has shifted to insecticide resistance [23] and to the characterization of germ line restricted chromosomes that occur in a few groups across the tree of life, including sciarids, cecidomyiids, and chironomids (reviewed in [24]).

*Pseudolycoriella hygida* (female adult in Figure 1) was originally described by Heni Sauaia and Maurílio A.R. Alves as a species of *Bradysia* Winnertz, 1867 [25] and recently moved to the genus *Pseudolycoriella* [26]. Rearing this species in the laboratory began with two females in 1965 in the Cellular Biology Laboratory of the Medical School of the University of São Paulo in Ribeirão Preto. The species was formally described with considerable detail based on adults [25], and the culture has been successfully maintained ever since. In 2016, the culture was transferred to the School of Pharmaceutical Sciences on the same university campus. Through the years, different laboratories in the state of São Paulo and Paraná have maintained *Psl. hygida* cultures with different degrees of success. Presently, this sciarid is cultivated in Ribeirão Preto, state of São Paulo (FCFRP-USP) and in Maringá, state of Paraná (UEM, State University of Maringá).

The life cycle of *Psl. hygida* lasts 41 days at 22 °C. Different diets, all based on partly decomposed leaves, have been employed over the years [27]. The use of diets with defined composition has been attempted. However, a decline in viability has been reported after some generations [28,29]. At present, the cultures of *Psl. hygida* are maintained on a diet of partly decomposed leaves of *Ilex paraguariensis* Saint Hilaire, 1822, which can be supplemented during the fourth larval instar with an artificial diet of defined composition [26,27].

The studies in Heni Sauaia’s laboratory in the 1970s and 1980s investigated the DNA and RNA puffs in the salivary gland chromosomes of *Psl. hygida*. By using a combination of cytological analysis and larval injection of drugs that either inhibit DNA synthesis or chemically modify DNA and radiolabeled precursors, his group confirmed that sciarid DNA puffs are sites of developmentally regulated gene amplification and abundant RNA synthesis [18,20,30,31,32]. Subsequent studies showed that the regulated synthesis of groups of salivary gland proteins was related to the expansion and regression of specific DNA puffs [33,34]. The observation that many of the DNA puff proteins were also found in the saliva of fourth instar larvae provided support for the idea that DNA amplification is a mechanism selected to provide a short-term boost in the amounts of those proteins that are destined to build the cocoon [33,34].

In the following years, *Psl. hygida* was employed as a model organism by different research groups both in Maringá and Ribeirão Preto. These studies included the characterization of the mitotic chromosomes and nucleolar organizing regions [35,36], the molecular characterization of DNA puff-forming regions (reviewed in [18,20]), the direct demonstration that DNA puff proteins are employed to build the cocoon [37,38], studies in transgenic *Drosophila*, as well as those that determined the levels of ecdysone titers and the role of the hormone on DNA puff gene expression at the end of the fourth larval instar (reviewed in [20]). Further studies established the mechanisms of programmed cell death in larval organs [18,22]. Together, these studies represent a consolidated contribution in the area of insect molecular biology and further emphasize the relevance of studying unconventional model organisms, such as sciarids.

The *Psl. hygida* genome has recently been sequenced, and an annotated version of the genome is available in the public databases [26]. The analysis of the *Psl. hygida* mitochondrial genome revealed that it contains all 13 protein coding genes, 22 tRNAs and 2 rRNAs characteristic of insect mitogenomes, and an unusually large control region of about 21 kb. A phylogenetic analysis based on Bibionomorpha mitogenome sequences reconstructed the phylogeny of the Bibionomorpha infraorder supported the monophyly of Sciaridae and revealed subfamily and genus-specific gene rearrangements [39]. The *Psl. hygida* genome is the third sciarid genome to be published and the first for the genus *Pseudolycoriella*, constituting a significant contribution to the field of Diptera genomics and to those working on sciarid biology [26]. However, some cytological and morphological aspects of *Psl. hygida* have not been previously published.

We present here an updated description of *Psl. hygida* adults and the first description of the morphology and behavior of immature stages, which were obtained from the long-term *Psl. hygida* laboratory culture. The identification and classification of the species based on morphological characters of the adults are presented together with a COI-based ML tree that reveals the position of *Psl. hygida* in the genus *Pseudolycoriella*. The description of the larval salivary gland polytene chromosomes was made more than 50 years ago [40], and photomicrographs of these chromosomes have been shown in different studies [32,41,42]. Here, we revisit the first description of these chromosomes and present a high-quality cytogenetical map that has remained unpublished since it was originally drawn. Additionally, we include a comparison between sciarid mitotic chromosomes displaying marks, such as centromere and nucleolar organizer. Taken together, these data extend the characterization of a sciarid species that has been employed as a model organism for over 55 years, and they provide a valuable resource for those studying the morphology, cytogenetics, and evolution of sciarids in particular and dipterans in general.

## 2. Materials and Methods

### 2.1. Cultivation and Life Cycle of Pseudolycoriella hygida

All specimens investigated in this work are derived from the culture currently maintained in the Monesi laboratory, located on the campus of the Universidade de São Paulo in Ribeirão Preto, Brazil. *Pseudolycoriella hygida* was maintained in plastic cultivation boxes (11 × 11 × 3.4 cm) layered with humid soil as previously described [26,34,43]. The life cycle lasts about 41 days at 22 °C [26], and although shorter durations for the life cycle (36 days) have been reported, the differences are mainly the result of cultivation at lower temperatures (20 °C) (for example, [34,41]) and/or not taking into account the 9-day-long embryonic development [43]. At 22 °C, the embryonic, larval, pupal, and adult stages last about 9 days, 22 days, 5 days, and 5 days, respectively.

The embryonic stage has been previously described [43], as well as the developmental stages of late fourth instar larvae, which is based on changes in the shape, size, and position of the eyespots [34]. The larval stage comprises three molts and four instars, and the larvae feed continuously on a diet of partially decomposed *Ilex paraguariensis* leaves. During the fourth larval stage, the diet can be supplemented with an artificial diet of defined composition (1.2% yeast extract, 1.4% cornstarch, 0.8% oatmeal flour, 1.2% agar). At the end of larval development, the larvae build individual cocoons in the soil. *Psl. hygida* is a digenic species, and the ratio between females and males is 5:1. The males emerge before the females, and copulation occurs as soon as the females emerge. Egg laying starts 24–48 h later. A female lays between 100 and 200 eggs [40,44]. Under laboratory conditions, no food is offered to the adults, which live for only 5 days. 

### 2.2. Molecular Methods and Techniques

#### 2.2.1. Preparation of Polytene Chromosome Spreads

Salivary glands were dissected and immersed in 7% perchloric acid for at least 2 h. The most anterior region was transferred to a drop of 50% acetic acid and squashed between the slide and coverslip. After removal of the excess acetic acid, the coverslips were sealed with paraffin. Photographs were taken under phase contrast with a Universal Microscope (Carl Zeiss GmbH, Jena, Germany).

#### 2.2.2. Preparation of Polytene Chromosome Maps

Larvae at the E5 eyespot stage were used for chromosome spreads as well as photographs. At this stage, most DNA puff primordia can already be observed. Several slides were inspected and photographed, including those from larvae at the same stage but from different cultivation boxes. To construct the map, photographs of the same chromosome were compared in order to rule out variations that could lead to map imprecisions.

#### 2.2.3. Preparation of Mitotic/Meiotic Chromosome Schemes

Several photographs displaying the whole chromosome complement of the six studied species were used. Chromosome lengths within the chromosome complement of the same species were measured, taking the largest autosome (A/IV) as a reference size (2.1–2.3 cm) for the other chromosomes within the same complement, and depicted proportionally in the chromosome schemes. The following data were employed for the scheme construction: *Pseudosciara pubescens* (Morgante, 1969) (as *Trichosia*) [45,46], *Rhynchosciara americana* (Wiedemann, 1821) [21,46,47], *Schwenckfeldina* sp. (as ‘*Schwenkfeldina*’; misspelling) [21,46], *Lycoriella ingenua* (Dufour, 1839) (as *Sciara pauciseta*) [48], *Bradysia tilicola* (Loew, 1850) (as *Sciara coprophila*) [49], and *Pseudolycoriella hygida* (Sauaia & Alves, 1968) (as *Bradysia*) [50].

#### 2.2.4. Molecular Analysis Based on COI Barcodes

The genomic DNA of all specimens with the sample ID beginning with ‘SDEI-Dipt’ was extracted at the Senckenberg Deutsches Entomologisches Institut (SDEI) using the EZNA Tissue DNA Kit (Omega Biotek, Norcross, GA, USA), following the manufacturer’s protocol. The extracted DNA was stored at −20 °C for later use. Typically, the entire specimen was used for DNA extraction. For the phylogenetic analyses, the mitochondrial region of the cytochrome oxidase subunit I gene (COI) was sequenced using the primers LCO_1490 and HCO_2198, which amplify the entire standard barcode region (658 bp) of the animal kingdom [51]. If these primers did not produce sufficient results, the primers LepF1 and LepR1 [52] or LCO_1490 and HCO_2198 in combination with the internal primers sym-C1-J-1718 [53] and C1-N1760 [54] were used. PCR products were purified and sequenced using an ABI3730XL sequencer with the Big Dye v. 3.1 Terminator Kit (Thermo Fisher Scientific, Schwerte, Germany) by Macrogen Europe (Amsterdam, The Netherlands). Sequences were manually edited using Geneious 9.1.2 [55] and aligned using BioEdit 7.2.5 [56].

The phylogenetic tree was constructed using the Maximum Likelihood (ML) method implemented in IQ-TREE 1.6.91 [57]. The best-fitting models for the analyses were determined using ModelFinder [58], which resulted in GTR+F+I+G4. The ML analysis included 1000 Ultrafast Bootstrap Approximation (UFBoot) replications using all sites. The tree was drawn to scale, with branch lengths corresponding to substitutions per site. The analysis included nucleotide sequences from 62 specimens (48 individuals of 44 *Pseudolycoriella* species and single individuals of 14 outgroup representatives), and the pairwise proportion of differences per nucleotide site (p-distance) between all sequences was calculated using MEGAX [59] with the pairwise-deletion option (Appendix A). All codon positions were included. UFBoot values are reported for 95% and above.

### 2.3. Morphological Methods and Techniques

#### 2.3.1. Slide-Mounting and Imaging of Immature Stages

Specimens of each larval stage were selected from the culture, together with male and female pupae. At least one specimen of each stage was permanently mounted on a microscope slide and deposited in the Museu de Zoologia, Universidade de São Paulo (MZUSP), São Paulo, Brazil. Specimens were cleared with KOH, dehydrated in ethanol, and mounted in Canada balsam (modified from [60]). The habitus and morphological details of the structures were imaged using light microscopy and photographed with a LEICA DC500 camera attached to a LEICA stereomicroscope model MZ-16 and to a compound microscope model LEICA DM2500 (all from Leica Microsystems GmbH, Wetzlar, Germany). Photos were stacked with Helicon Focus 6. Images were edited with Adobe Photoshop CC, and plates were prepared with Adobe Illustrator CC.

#### 2.3.2. Slide-Mounting and Imaging of Adults

A stereo zoom microscope OLYMPUS SZX12 in combination with a ring light system OLYMPUS KL 2500 LCD was used for the preparation of adults. All sciarid specimens were initially preserved in 70% ethanol. They were first transferred to 96% ethanol and then subjected to a creosote treatment (approx. 15 min per step). The creosote step allowed residual dehydration of imagines and lightening of very dark, heavily sclerotized body parts. After dehydration, the adults were placed in lateral view on slides with a liquid Canada balsam/xylene mixture. Females were positioned under a coverslip without prior dissection of body parts (Figure 1). For males, genitalia were separated under permanent microscopic observation with two clamped insect needles (size 1) and embedded in the ventral view next to the body [61]. A HERAEUS Instruments T6060 oven with a constant operating temperature of 50 °C was used to dry the permanent slides over a period of two to three weeks.

A fully automated LEICA DM6 B microscope system was used in the morphological examination and measurement of adult specimens. The microscope was equipped with HC PLAN eyepieces (10×), seven objective lenses (1.25×, 5×, 10×, 20×, 40×, 63×, 100×), an intermediate magnification changer (1.0×, 1.25×, 1.6×), and an LAS interactive measurement module. Images were documented with a LEICA DMC6200 digital camera mounted on the LEICA DM6 B microscope (both Leica Microsystems GmbH, Wetzlar, Germany). Between 10 and 40 images were taken at different focal planes, stacked, and fully automatically computed into a depth-sharp photo with the LAS Montage MultiFocus software, version 4.13.0. Images were edited with Zooner Photo Studio, version 10, and arranged as plates with Adobe Photoshop CS6, version 13.0. After completion of the morphological studies, 44 permanent slides of 62 *Psl. hygida* adults were deposited in the Diptera collection of the Senckenberg German Entomological Institute (SDEI) in Müncheberg, Germany (for details see Section 3.4).

### 2.4. Species Identification and Classification

The identification of species was carried out according to the keys, descriptions, and figures by several authors [1,3,4,5,7,25,61,62,63,64,65,66,67,68,69,70,71,72,73,74,75,76,77,78,79,80]. The morphological terminology follows [7,70,81], with some minor adaptations to abbreviations (see Section 2.5, ‘Morphological abbreviations’).

### 2.5. Abbreviations

The following abbreviations and terms are used in the text:

Genetic abbreviations and explanation of terms: **BIN** = Barcode Index Number; **BOLD** = Barcode of Life Data System; **COI** = Mitochondrial cytochrome oxidase I; **DNA** = Deoxyribonucleic acid; **N/A** = not available; **NORs** = Nucleolar Organizer Regions.

Morphological abbreviations and explanation of terms: *Larvae*. **ant** = antenna; **as** = abdominal spiracle; **cl** = clypeus; **clp** = clypeus medial process; **crd** = cardo; **E1** = eyespot pattern 1; **E5** = eyespot pattern 5; **E7** = eyespot pattern 7; **el** = ecdysial line; **frp** = frontal plate; **gl** = galea; **gn** = gena; **hyp** = hypopharynx; **lb** = labrum; **mdb** = mandible; **mxl** = maxilla; **p1**–**p9** = sensory pit 1–9; **poc** = postoccipital carina; **st** = stipes; **tas** = terminal abdominal segment; **ths** = thorax spiracle. *Pupae*. **abd seg** = abdominal segment; **ant sh** = antennal sheath; **ce sh** = cercus sheath; **gs sh** = gonostylus sheath; **lg sh** = leg sheath; **pl sh** = palpal sheath; **prn** = pronotum; **prs** = prothoracic spiracle; **sp** = abdominal spiracle; **vt** = vertexal tubercle; **vt pl** = vertexal plate; **wg sh** = wing sheath. *Adults*. **bM** = basal part of Media; **C** = costal vein; **c** = distance between apex of R_5_ and end of C; **CuA_1_** = first (anterior) branch of cubital vein; **CuA_2_** = second (posterior) branch of cubital vein; **c/w** = length ratio between c and w; **H/K index** = ratio between halter length (H) and halter knob length (K); **M_1_** = first branch of media; **M_2_** = second branch of media; **M-fork** = medial vein fork; **p_1_** = fore leg(s); **p_2_** = middle leg(s); **p_3_** = hind leg(s); **R** = radius, or radial vein; **R_1_** = first (anterior) branch of radius; **R_5_** = third branch of radius; **r-m** = Radius/Media sector (r-m) as basal part of R_5_; **stM** = stem of medial veins M_1_ and M_2_; **stCuA** = stem of cubital veins CuA_1_ and CuA_2_; **w** = distance between apex of R_5_ and apex of M_1_.

Nomenclatural abbreviations: ***B.*** = *Bradysia*; ***Psl.*** = *Pseudolycoriella*; ***Pss.*** = *Pseudosciara*; ***Rh.*** = *Rhynchosciara*; ***L.*** = *Lycoriella*; ***S.*** = *Sciara*; ***T.*** = *Trichosia*.

International country codes: **ARG** = Argentina; **BRA** = Brazil; **CAN** = Canada; **FIN** = Finland; **GER** = Germany; **KOR** = South Korea; **NOR** = Norway; **NZL** = New Zealand; **SEY** = Seychelles; **USA** = United States.

Collection codes: **BFCO** = BioFokus, Oslo, Norway; **CBGC** = University of Guelph, Centre for Biodiversity Genomics, Guelph, Canada; **MACN** = Museo Argentino de Ciencias Naturales ‘Bernardino Rivadavia’, Buenos Aires, Argentina; **MZUSP** = Museu de Zoologia, Universidade de São Paulo, São Paulo, Brazil; **NHMO** = University of Oslo, Natural History Museum, Oslo, Norway; **NTNU** = Norwegian University of Science and Technology, University Museum, Trondheim, Norway; **NZAC** = New Zealand Arthropod Collection, Auckland, New Zealand; **SDEI** = Senckenberg Deutsches Entomologisches Institut, Müncheberg, Germany; **SNUC** = Seoul National University, College of Agriculture and Life Sciences, Seoul, South Korea; **UOLC** = University of Ostrava, Lab Reference Collection, Ostrava, Czech Republic; **ZFMK** = Zoologisches Forschungsmuseum Alexander Koenig, Bonn, Germany; **ZSMC** = Zoologische Staatssammlung des Bayerischen Staates, Munich, Germany.

## 3. Results

### 3.1. Pseudolycoriella hygida in the Literature

In the past 55 years, *Pseudolycoriella hygida* has either itself been the subject of scientific investigations or the species has been used for comparison with other study objects. Our literature search retrieved 71 publications in which the name ‘*hygida* Sauaia & Alves’ occurs. Of these, 68 were in cytogenetics and/or molecular biology, and 2 were in taxonomy and systematics. Only Trinca et al. [26] took an interdisciplinary approach by linking research results from genetics and systematics. In the following overview, the cited publications that appeared between 1968 and 2023 are listed chronologically, because the results of one study generally form the basis of the next. The authors are sorted alphabetically within a given publication year.

Literature in systematics: *Bradysia hygida* Sauaia & Alves—[25]: 85, plate 1, Figures 1–6, plate 2, Figures 7–13 (original description); [82]: 58 (catalogue of Neotropical sciarids). *Pseudolycoriella hygida* (Sauaia & Alves)—[26]: 20, Figure 1A (re-classification).

Literature in cytogenetics and/or molecular biology: *Bradysia hygida* Sauaia & Alves—[40]: 4, Figures 1–21; [32]: 130, Figures 1–8; [31]: 2, Figures 2–10; [83]: 153; [84]: 1311, Figures 1–5; [34]: 280, Figures 1–5; [28]: 383; [29]: 272; [85]: 247, Figures 1–7; [15]: 235; [86]: 439, Figures 1–4; [87]: 777, Figures 1–3; [27]: 344; [88]: 121, Figures 1–4; [89]: 715, Figures 1–9; [90]: 10, Figures 1–4, 9–20; [16]: 307; [33]: 606, Figures 1–3; [91]: 167; [92]: 463; [93]: 559; [37]: 67, Figures 1–3, 5–6; [35]: 101, Figures 1–4; [44]: 167, Figures 1 and 2; [94]: 541, Figures 1–4; [22]: 1982, Figures 1–8; [95]: 851, Figures 1–4; [41]: 16, Figures 1–4; [96]: 737, Figures 1–8; [36]: 57, Figures 1–3; [97]: 348, Figures 1–4; [98]: 145; [38]: 531, Figures 1–7; [99]: 254, Figures 1–4; [100]: 173, Figure 6; [101]: 23, Figures 1–6; [102]: 744; [103]: 352, Figures 1 and 2; [104]: 4, Figures 11 and 12C–D (in part misspelled as ‘*higida*’; correctly *hygida*); [105]: 8; [106]: unpaginated; [42]: 609, Figures 1, 2 and 3D, 4; [107]: 43; [20]: 166, Figures 6.1 and 6.4; [108]: 848, Figures 1–7; [109]: 438; [30]: 1143, Figures 1–5; [110]: unpaginated, Figure 2; [111]: 1186; [112]: 3, Figure 2B; [113]: 2; [114]: 2062, Figures 1–4; [50]: 2178, Figures 1–4; [115]: 287, Figure 12.6; [116]: 338; [117]: 31, Figures 1–5; [118]: 598; [119]: 319; [18]: 361, Figure 2; [120]: 1, Figures 2, 3 and 5; [121]: 2; [122]: 5, Figures 1–9; [43]: 270, Figures 1–4; [22]: 1982, Figures 1–8; [21]: 8; [123]: unpaginated, Figure 1; [124]: 20, Figures 2.1 and 14; [39]: 483, Figures 1–5, S1, S4, S5 and S7. *Pseudolycoriella hygida* (Sauaia & Alves)—[26]: 2, Figures 1–4, 5A and S1–S10.

### 3.2. Development and Behavior of Immatures and Adults

At 22 °C, first instar larvae hatched from eggs between the 9th and 10th day of embryonic development, already showing a sclerotized cephalic capsule. First instar larvae began to feed immediately after hatching, and the gut darkened due to the presence of ingested matter (Figure 2A,B). During the whole larval stage, the larvae secreted saliva onto the surface of the food, comprising partly decomposed *Ilex paraguariensis* leaves, and the larval groups spread out over the substrate. As the larvae began to move around, additional substrate was placed in the box. Larvae moved in groups, as is common in sciarids, from the older food that was exposed to saliva to the fresher food. Second instar larvae molted on the fourth day of larval development, staying close to the bottom of the box. Second instar larvae could be observed with or without the sclerotized cephalic capsule during this transition, but all larvae usually had a sclerotized cephalic capsule by the fifth day. Third instar larvae emerged on the eighth day, with a new replacement of the cephalic capsule. During the third instar, the larval gut became even darker.

The molt of the fourth instar larva occurred between the 11th and 12th day of development and was accompanied by another change of the cephalic capsule. The food of fourth instar larvae was augmented with an artificial supplement of defined composition (for details, see Section 2.1). Within 18 days, larvae at stage E1 (eyespot pattern 1) could be seen [34], while the larvae began to feed mainly on the artificial diet at stage E5. From stage E5 onwards, the artificial diet was added to the edges of the leaf-based food and on the soil, which induced the larvae to migrate towards the soil. After stage E7, larvae buried themselves in the soil, adopted a half-moon shape, stopped moving, and started to build individual cocoons. At the beginning of pupation, the fourth instar integument moved posteriorly, and the cephalic capsule remained attached at the posterior end of the pupa (Figure 2C). Adults emerged mostly on the fifth day after pupation. Males usually emerged before females and moved around the female pupae. As soon as the females emerged, copulation occurred. Females started laying eggs 24–48 h after emerging. Adults lived for about 5 days [26].

### 3.3. Description of Immature Stages

**Larva** (Figure 2A,B, Figure 3 and Figure 4A–D). White, head capsule shiny black, larva cylindrical, eucephalous, semi-translucent; no prolegs, but terminal abdominal segment (tas) slightly lobate, helping in locomotion. First instar metapneustic (Figure 2A,B); second and third instars propneustic; fourth instar hemipneustic, with body with 12 segments (Figure 3A–D). Frontal plate (frp) typically triangular with the genae at each side, posterior end not very slender; nine sensory pits (p1–p9), six on the frontal plate, three along the lateral ecdysial lines (el) (Figure 4A). Posterior tentorial bridge complete; postoccipital carinae (poc) medially ending close together, but not touching (Figure 4A). Labrum (lb) membranous, with a pair of lobes (Figure 4B). Clypeus with a distinct medial process (clp) projecting anteriorly (Figure 3C and Figure 4B). Genae (gn) meeting ventrally at two points; a wide membranous genal foramen medially, with trapezoid shape, blunt at anterior end. Antenna (ant) small, arrow-shaped (Figure 4B), lying above the base of the mandible. Eye lightly pigmented below antennal base. Premandible not strongly sclerotized. Mandible (mdb) not strongly sclerotized, teeth anteromedial, prostheca well-defined (Figure 4D); maxilla (mxl) with several progressively stronger teeth, most distinct medialIy (Figure 3D and Figure 4C); base of maxilla with lateral and proximal rods; maxillary palpus with wide stipes, cardo triangular; hypopharynx (hyp) rod V-shaped, between bases of maxillae. Cardo (crd) trapezoid, latero-ventral margin curved (Figure 4B). No sclerotized spicules on abdominal creeping welts. Abdominal segments 1–7 with small ventral creeping welts, bearing rows of fine spicules, but no special lobes, projections, or setae (Figure 3A,B). Fourth instar with one pair of spiracles on prothorax (ths), each spiracle with two openings, abdominal segments (as1, as2) 1 to 7 with one pair of spiracles, each spiracle with a single opening, segment 8 with no spiracles. Measurements (end of the fourth instar): male body = 6.5 mm × 0.9 mm; female body = 8.6 mm × 1.0 mm.

**Pupa** (Figure 2C, Figure 5A–D and Figure 6A–D). Head with vertexal plate (vt pl) and dorsal surfaces of antennal sheaths (ant sh) visible ventrally (Figure 5C); vertexal plate situated between narrow pronotum and two vertexal tubercles (vt) (Figure 5A and Figure 6A); tubercles each bearing a strong bristle; antennal sheath (ant sh) extending obliquely to and along anterior margin of wing sheath (Figure 5B). The wing (wg sh) (Figure 5C), antennal (ant sh) (Figure 5B and Figure 6B), and leg sheaths (lg sh) (Figure 5C) pressed against the body and directed towards the abdomen. Frontoclypeal apotome situated between base of antennal sheath and proboscial sheath, antennal sheath and proboscial sheath, with posterior margin extending tongue-like over anterior edge of broadly rectangular proboscial sheath. Palpal sheath (pl sh) extending laterally below compound eyes (Figure 5C). Thorax not strongly arched; narrow pronotum (prn) extending from anterior spiracular tubercles (prs) (Figure 6B); mesonotum featureless; metanotum lying between halter sheaths (Figure 5B,D). Wing sheath extending posteroventrally to posterior margin of abdominal segment 2 (abd seg 2) (Figure 5B–D). Legs closely appressed and folded (Figure 5C,D). Anterior spiracle on a blunt tubercle, with 3–13 spiracular openings arranged in a semicircle or in a line. Abdomen with nine segments (abd sg), sometimes covered with very short wart-like spines that are denser on tergites and on terminalia than elsewhere (Figure 5A–D). Abdominal spiracles (sp) present on segments 2–7; each spiracle slightly stalked, with a single opening (Figure 5B–D). Terminal segment of male bearing a sheath enclosing each gonostylus (gs sh) and a separate sheath enclosing each cercus (ce sh); tergite 9 sometimes with two posterolateral prolongations covered with short spines (Figure 6C,D). Measurements (Figure 5): male body = 3.0 mm × 0.9 mm; female body = 3.8 mm × 1.2 mm.

### 3.4. Redescription of Adults and Distribution of Pseudolycoriella hygida

**Type material.** The type series mentioned the male holotype and 49 paratypes (28 males, 21 females), Brazil, ‘Campus of Medical School of Ribeiro Prêto (Fazenda Monte Alegre)’ (State of São Paulo, Ribeirão Preto, campus of the Universidade de São Paulo, Faculdade de Medicina de Ribeirão Preto, Fazenda Monte Alegre), without detailed collection data. MZUSP holds the male holotype, three male and three female paratypes, all slide-mounted. All type specimens were reared in a laboratory culture, originally traced back to two egg-laying females caught in the grass on the campus grounds.

**Additional material.** Brazil, 23 males (one of them with sample ID SDEI-Dipt-0002256), 39 females (one of them with sample ID SDEI-Dipt-0002255), Brazil, State of São Paulo, Ribeirão Preto, Vila Monte Alegre, Avenida do Café, Campus da Universidade de São Paulo (USP), Faculdade de Ciências Farmacêuticas de Ribeirão Preto, Departamento de Análises Clínicas, Toxicológicas e Bromatológicas, Laboratório de Biologia e Genômica de Diptera (Bloco S), -21.1694722°N -47.8488611°E (21°10′10.10″S 47°50′55.90″W), 580 m, laboratory culture (breeding), 07.06.2022, leg. Nadia Monesi (all in SDEI).

**Additional records taken from BOLD.** Argentina, one female (sample ID BIOUG23471-G01), Formosa province, Reserva El Bagual, -26.303°N -58.815°E (26°18′10.80″S 58°48′53.99″W), 57 m, 09.08.2013, leg. Pablo Tubaro (in MACN); one specimen of unknown sex (sample ID BIOUG23277-A10), same locality, -26.303°N -58.815°E (26°18′10.80″S 58°48′53.99″W), 57 m, 28.06.2013, leg. Pablo Tubaro (in MACN); one specimen of unknown sex (sample ID BIOUG23645-D09), same locality, -26.3028°N -58.8150°E (26°18′10.08″S 58°48′53.99″W), 57 m, 11.10.2013, leg. Pablo Tubaro (in MACN).

**Distribution.** Only known from the Neotropical realm: Argentina (new record); Brazil.

**Redescription.** *Male* (Figure 7B, Figure 8A,D and Figure 9A–C). *Head.* Head capsule rounded; all ocelli present; prefrons and clypeus short, both with short and strong setae. Eye bridge closed, with 2–3 rows of ommatidia. Antennae short and dark brown, consisting of scape, pedicel, and 14 flagellomeres; antennal base dark brown; scape bowl-shaped, pedicel roundish; flagellomeres (Figure 8A) with sharp edge between basal portion and neck; necks short and monotonous brown; basal portions coarsely, densely and protrudingly setose; surface of basal portions rough, partially with honeycomb-like structure; fourth flagellomere (Figure 8A) 1.60–1.90 times as long as wide; neck only 0.18–0.23 times as long as the basal portion; basal portion with inconspicuously short, fine sensilla; setae about 2/3 times as long as the basal portion wide. Palps long, 3-segmented; all palpal segments rod-shaped and almost equal in length; first palpal segment 1.15–1.38 times as long as segment 2, and 0.86–1.07 times as long as segment 3; third palpal segment 0.85–1.34 times as long as segment 2; first palpal segment somewhat thickened and with 4–8 long setae, one of them distinctly longer; sensory area simple, unmodified (without deepened sensory pit); sensilla short and fine; all setae on palpal segments 2 and 3 short. *Thorax.* Dark brown to black; thoracic sclerites not fused. Anterior pronotum with 5–8 setae. Postpronotum non-setose. Scutum weakly arched, dorsal marginal areas blackened, lateral and central areas densely covered with coarse, long bristles interspersed with fine, dark setae (these about 2/5 to 1/3 times as long as the coarse bristles). Scutellum well-developed, with 4 long, strong setae dorsally and some shorter, weaker setae ventro-laterally. Mediotergite and metanotum non-setose. Katepisternum high and triangular, centre of dorsal third sometimes with a strong, short seta (n = 4). Coxae and legs robust, dark brown; coxae of p_1_ caudally lightened to yellowish; coxae, femora and tibiae robust, neither conspicuously shortened nor elongated, with dark brown to black setae; all tibiae with long, stylet-shaped spurs; tibiae of p_2_ and p_3_ with two spurs of equal length; tibiae of p_1_ without spinose setae in the dense setosity; apex of fore tibia (as in Figure 8C) with a small anteroapical patch with flat curved margin, consisting of 6–11 setae, which tend to form an irregular row (this tibial organ narrow, about 1/3 times as wide as the apex of fore tibia); hind tibiae with posterodorsal row of spinose setae, these quite long and irregularly arranged in a weakly developed row; apex of hind tibia only with 9–15 irregularly arranged spinose setae, without closed row. Claws toothed, with 4–5 teeth on the inner side (2 coarse and 2–3 much finer). Wings (Figure 7B) fumose, brownish; anal area well-developed; wing membrane without macrotrichia; R/R_1_ vein complex short, R_1_ merged with C well before the base of M-fork; R_1_ = 0.69–0.75 R, both dorsally setose only; R_5_ with macrotrichia dorsally and ventrally in distal 1/5 to 2/5 (Figure 8D); posterior wing veins very strong, only stM and anal veins weakly developed; CuA_1_, CuA_2_, stCuA and M_2_ without macrotrichia; M_1_ mostly with one to 11 macrotrichia (Figure 8D), rarely bare; stM mostly bare, very rarely in the distal half with one to 3 macrotrichia; M-fork distinctly shorter than stM (about 0.70 times as long as stM); bM = 1.13–1.32 r-m; bM bare, r-m completely covered with macrotrichia; stCuA = 0.81–0.92 bM; c = 0.53–0.58 w. Halter dark, short-stalked; H/K index = 1.95–2.07; knob of halter with 2 rows of short setae.

*Abdomen*. Dark brown, laterally densely covered with dark microtrichia (7–9 microtrichia arranged in roundish groups); all tergites and sternites with coarse, long, dark brown to black setae; tergites 5–8 well developed, without lobus-like extensions or conspicuous groups of setae. Hypopygium (Figure 9A) slightly wider than high, dark brown; ventral base of hypopygium without intercoxal lobus or group(s) of setae; gonocoxae compact, only with a strong, long seta ventroapically on each side; ventral emargination of gonocoxae V-shaped and clothed in short setae, bare or with few setae on the upper third. Tergite 9 trapezoidal, about 1.30 times higher than at its base wide. Gonostylus (Figure 9B) slender, elongate-oval, 2.60–2.80 times as long as wide; apex of gonostylus angled and rounded, densely and coarsely setose, without tooth apically; inner margin of gonostylus slightly bulbous in the centre, slightly concave above up to the apex; upper third of gonostylus with a moderately long, downcurved whiplash seta, above with 3–4 slender megaseta (arrangement of mesial megaseta often 2+1, very rarely 1+1+1 or 2+2); all megaseta longer than the dense setosity of gonostylar apex; middle third of gonostylus with 2–4 long setae inserting dorsally and only 1/4 shorter than the stronger whiplash seta. Tegmen (Figure 9C) about as high as wide, laterally without processes; apex of tegmen membranous, convex rounded; basolateral apodemes very short, sclerotised; finger-like process present (Figure 9C), this vertical structure long, reaching from the middle to the apex of tegmen. Aedeagus complex strongly sclerotized, with a very high and narrow funnel-shaped base; ejaculatory apodeme very short and strong; area of aedeagal teeth very narrow, band-shaped; all teeth short, very fine and single-pointed. Measurements: body length = 2.46–3.03 mm; wing length = 2.15–2.61 mm.

*Female* (Figure 1, Figure 7A and Figure 8B–C). Habitus and coloration as in Figure 1. *Head*. Antennae shorter than in male; fourth flagellomere (Figure 8B) 1.45–1.50 times as long as wide; basal portion compact, with fewer setae and more pronounced honeycomb-like surface structure; neck of fourth flagellomere 0.18–0.20 times as long as basal portion.

*Thorax*. Scutellum with at least 4 strong, long setae, but often with 6–8, which hardly differ in length. Katepisternum mostly bare, sometimes with 1–2 short setae in the centre of the dorsal third (n = 3). Anteroapical tibial organ on the fore tibia, as in Figure 8C. Wings (Figure 7A) distinctly larger and somewhat wider than in male; anal area well-developed, but less arched; R/R_1_ vein complex short, R_1_ = 0.65–0.74 R; R_5_ longer as in male, with macrotrichia dorsally and ventrally in distal 1/5 to 2/5; posterior wing veins somewhat darker; M-fork and stM longer, M-fork 0.76–0.81 times as long as stM; M_1_ mostly with one to 12 macrotrichia, rarely bare (n = 2); stM mostly bare, very rarely in the distal half with one to 5 macrotrichia; bM = 1.02–1.33 r-m; stCuA = 0.76–0.93 bM. Halter with shorter stalk; H/K index = 1.75–1.92. All other characters as in male, including the coloration and setosity of *abdomen*. Measurements: body length = 3.55–4.07 mm; wing length = 2.65–3.05 mm.

### 3.5. Identification and Classification of Pseudolycoriella hygida Based on Morphological Characters of Adults

Of the 161 described *Pseudolycoriella* species, 135 species have been classified into 10 species groups based on the morphological characters of male adults [3,61,63,68,71] as follows: *Psl. aculeacera* group (5 species; erroneously also referred to as ‘*Psl. ovistyla* group’ for the same species group in [71]); *Psl. bruckii* group (74 species); *Psl. horribilis* group (2 species); *Psl. longicostalis* group (11 species); *Psl. macrotegmenta* group (14 species; formerly also referred to as ‘*macrotegmenta* complex’); *Psl. morenae* group (2 species); *Psl. nodulosa* group (6 species; formerly also referred to as ‘*nodulosa* complex’ or subgenus *Ostroverkhovana* Komarova); *Psl. quadrispinosa* group (9 species); *Psl. triacanthula* group (8 species); and *Psl. torva* group (4 species). 

No adequate species groups have yet been defined for the remaining 26 *Pseudolycoriella* species (16.15%) with deviating body and genital characters. They are currently still regarded as ‘unplaced’ within *Pseudolycoriella*. This ‘unplaced species mix’ also includes *Psl. hygida* (Sauaia & Alves, 1968) as well as *Psl. coecoalata* Mohrig, 2003 described from Costa Rica, *Psl. florentissima* Mohrig & Rulik, 2004 from Dominican Republic, *Psl. nocturna* Mohrig & Kauschke, 2019 from Canada, *Psl. skusei* Mohrig, Kauschke & Broadley, 2016 from Australia, *Psl. senticosa* Vilkamaa, Hippa & Mohrig, 2012, and *Psl. tribulosa* Vilkamaa, Hippa & Mohrig, 2012, both only known from New Caledonia. The males of these seven species have long, slender gonostyli with usually 3–4 mesial megasetae and at least one downcurved whiplash seta on the upper third of gonostylus, whereby intercoxal differentiations at the ventral base of gonocoxae are absent. *Pseudolycoriella nocturna* and *Psl. skusei* were excluded from further analysis because both species lack the curved margin of the anteroapical tibial organ. In addition, *Psl. skusei* has an unusually high, strongly narrowed tegmen with a very long aedeagus, and *Psl. nocturna* has untoothed claws and some setae on the mediotergite. *Pseudolycoriella florentissima* can also be ruled out, because in this species the tegmen has no finger-shaped median process and the gonostylar megasetae are arranged subapically in a narrow group, similarly to representatives of the *Psl. bruckii* species group. In the remaining species *Psl. hygida*, *Psl. coecoalata*, *Psl. senticosa,* and *Psl. tribulosa*, the mesial megasetae on the upper half of gonostylus are arranged one below the other, usually clearly separated from each other. These species can be divided into two morphotypes according to the coloration of the basal antennal flagellomeres, the surface structure on the basal portion of flagellomeres, the shape of gonostyli, the length of setae on medial gonostylar margin, the length and position of whiplash seta, and some structures of the aedeagus complex.

Morphotype A (*Psl. senticosa*, *Psl. tribulosa*): Flagellomeres 1 and 2 yellow; basal portions of flagellomeres without conspicuous surface structures; gonostylus club-shaped, constricted at base; gonostylar apex not elongated, with linear inside edge; inner margin of gonostylus with setae moderately long, distinctly shorter than the whiplash seta; whiplash seta very long, inserted subapically close to the apex of gonostylus; base of aedeagus short, wide, funnel-shaped, and membranous; area of aedeagal teeth roundish, small.

Morphotype B (*Psl. hygida*, *Psl. coecoalata*): Flagellomeres 1 and 2 dark brown; basal portions of flagellomeres with pit-shaped, partially honeycomb-like connected surface structure; gonostylus equally wide from base to apex, not constricted at base; gonostylar apex elongated, with angled inside; inner margin of gonostylus with setae long, only slightly shorter than the whiplash seta; whiplash seta distinctly shorter, inserted medially at the base of upper gonostylar third; base of aedeagus long, narrow, funnel-shaped, and sclerotized; area of aedeagal teeth band-shaped, narrow.

Based on the morphological analysis carried out above, *Psl. hygida* (Sauaia & Alves) is closely related to *Psl. coecoalata* Mohrig, 2003 from Costa Rica, which was figured in detail by Mohrig [67] (compare with Figure 26a–f). The male of *Psl. hygida* differs from this species by its shorter flagellomeres (index of fourth flagellomere = 1.6–1.9), a shorter gonostylus with a wider apex and 3–4 mesial megaseta, a shorter tegmen (only as high as wide at the base) with a long finger-like process, and a short ejaculatory apodeme. In contrast, in *Psl. coecoalata,* the male flagellomeres are longer (index of fourth flagellomere = 2.4), the gonostylus is longer, with a narrower apex and only 2 megaseta on the medial side of the gonostylus, and the tegmen is significantly higher than wide. In addition, in *Psl. coecoalata,* the finger-like median process on the tegmen is only half as long and the strong ejaculatory apodeme of aedeagus twice as long as in *Psl. hygida*. Further differences exist in the setosity of some wing veins, which can be used for the identification of *Psl. hygida* and *Psl. coecoalata* in both sexes. The r-m vein is completely covered with macrotrichia in *Psl. hygida* (bare in *Psl. coecoalata*), and the distal area of R_5_ is also covered dorsally with macrotrichia (in *Psl. coecoalata* with ventral macrotrichia only). In addition, veins M_1_ and stM in *Psl. hygida* often retain some macrotrichia (atavism), which has not yet been observed in *Psl. coecoalata*.

### 3.6. Genetic Identification and Relationship to Other COI-Analyzed Pseudolycoriella

To identify the genetic relationship of *Psl. hygida* to other *Pseudolycoriella* species and to evaluate genetic variability, we conducted a phylogenetic analysis using COI barcode sequences. The study presented here included 44 described *Pseudolycoriella* species of which the COI barcodes are currently known (Figure 10, Table 1). In order to test the monophyly of *Pseudolycoriella*, 13 genus representatives of Sciaridae were selected that are currently either considered to belong to the ‘*Pseudolycoriella* genus group’ or may be close to this complex based on morphological structures, especially due to the presence of very long, mesial, or whiplash gonostylar setae [1,5].

Among them are specimens of 10 type species, representatives of 2 species groups postulated in Menzel and Mohrig [61] that differ morphologically from the type species of genera, and 1 available *Spinopygina* species (Table 1). The mycetophilid family Diadocidiidae Winnertz, 1863 was selected as the outgroup of Sciaridae, with the type genus *Diadocidia* Ruthe, 1831 and the type species *Diadocidia ferruginosa* (Meigen, 1830).

Our analysis revealed that none of the three COI sequences of *Psl. hygida* from Argentina downloaded from BOLD (sample IDs BIOUG23645-D09, BIOUG23277-A10, BIOUG23471-G01; identified as ‘Sciaridae sp.’) nor the two barcodes of *Psl. hygida* from Brazil (sample IDs SDEI-Dipt-0002255, SDEI-Dipt-0002256) showed any intraspecific variability (0.00%) within each population. However, sequences of the specimens from Argentina diverged by 0.46% from those from Brazil. We attribute this intraspecific variability to the geographic distance between these populations. The COI analysis also showed that the five specimens from Argentina and Brazil form a monophylum and that this species is a representative of the genus *Pseudolycoriella* (Figure 10). The nearest-neighbor species to *Psl. hygida* in our analysis was *Psl. bruckii* (Winnertz, 1867), with an interspecific distance of 16.64% (Appendix A).

Molecular data for *Psl. hygida* (partly published under the names ‘*Bradysia hygida*’ or ‘Sciaridae sp.’) are deposited in BOLD Systems and GenBank as follows:

***COI barcodes***: BIN ID BOLD:ACW7953 [125], and this study.

***Ribosomal RNA***: 5S, GenBank accession number AY224134 [99]; 18S, GenBank accession number JQ652461 [50]; 28S, GenBank accession number JQ652462 [50].

***Mitogenome***: GenBank accession number MW442371 [39].

***Genome assembly***: GenBank accession number WJQU00000000, BioSample SAMN12911131 [26].

### 3.7. Cytogenetics

The polytene chromosome complement of *Psl. hygida* follows the pattern documented for other sciarid species that have been cytogenetically studied: there are three autosomal chromosomes named A, B, and C according to Metz [126] or IV, III, and II, respectively, according to Crouse and Smith-Stocking [127], in addition to the X chromosome (Figure 11). The latter is a pair in the female soma while it is only single in the male soma [35,36]. Such a general feature of all of the sciarid flies allows the identification of the male polytene X chromosome as it stains weaker than the female polytene X that originates from two chromatids instead of one. Polytene chromosome maps of *Psl. hygida*, although constructed half a century ago, are shown for the first time in this report (Figure 12) and have since been useful for the identification of Nucleolar Organizer Regions (NORs) and DNA and RNA puffs in addition to other transcriptionally active sites that do not develop large puffs (reviewed in [18,20]).

Except for *Hybosciara fragilis* (Morgante, 1969), which has never been recollected since the 1960s, sciarid flies that have been exploited in cytogenetics and/or molecular biology are restricted to six genera, namely *Bradysia* [49,128], *Rhynchosciara* [21,46,47], *Lycoriella* [48], *Pseudolycoriella* [26,39], *Pseudosciara* [45,46], and *Schwenckfeldina* [21,46]. With the aim of finding clues regarding whether and how chromosome complements of sciarids are conserved or diverge during genus evolution, we compared representatives of these genera using data available in the literature. Their basic chromosomal features, such as the localization of centromeres and NORs, are shown schematically (Figure 13).

Three conserved chromosome marks emerge from Figure 13. All included species display a single NOR within each chromosome complement that is always close to the centromere. In addition, in the studied species the large autosome (A or IV) is always metacentric or sub-metacentric (Figure 13). Three sub-groups can be formed based on NOR location: in *Pseudosciara* and *Lycoriella,* it is located in the metacentric or sub-metacentric X chromosome; in *Rhynchosciara* and *Bradysia,* it is located sub-telocentrically in chromosome X; and in *Schwenckfeldina* and *Pseudolycoriella,* the NOR is sub-telocentric in the shortest autosome (chromosome C or II).

## 4. Discussion

### 4.1. Breeding of Pseudolycoriella hygida and Morphology of Immature Stages

*Pseudolycoriella hygida* larvae are maintained on a diet of partially decomposed *Ilex paraguariensis* leaves, but it remains to be determined if they feed exclusively on decomposing vegetable matter or also on the associated fungi and/or microorganisms that participate in the decomposition process. *Pseudolycoriella hygida* larvae are gregarious, and during the larval stage they migrate towards the fresher food. Nevertheless, the larvae will continue to feed on older food until it turns into slurry if fresher food is not added to the culture. Attempts to feed *Psl. hygida* larvae on living plants were not successful, whereas larvae fed solely on white mushrooms (*Agaricus bisporus*) did not show alterations in the life cycle (Marques and Monesi, unpublished observations). Different sciarid species can use different food sources [5,130], and even though the feeding behavior under laboratory conditions may differ from that which occurs in nature, we propose that further studies should be carried out on *Psl. hygida* to better understand larval feeding habits and behavior in the very species-rich and widespread genus *Pseudolycoriella*. 

A laboratory culture is a valuable resource in the study of different traits of an insect species, such as the life cycle, development, feeding habits, and molecular mechanisms. Sciarid larvae found in the field are difficult to associate with adults, and the larvae and pupae of relatively few species have been described. Menzel and Mohrig [61] pointed out that preimaginal stages of less than 40 sciarid representatives have been described morphologically so far. Most of these are black fungus gnats that have drawn attention either as pests or because they undertake large larval migrations, a natural phenomenon popularly called ‘armyworm’ or ‘snakeworm’ [61].

General morphology, terminology, and illustrations can be found mainly in [6,61,70,131,132,133,134,135,136,137]. The presence of a genal foramen, the membranous medial area ventrally, seems to be a derived feature shared by all sciarids—also seen, for example, in *Corynoptera* [134] and *Scythropochroa* [136], although with some variation in shape and size. A medial sclerotized projection of the clypeus, also seen in *Scythropochroa* [136], is a derived feature probably not universally present in sciarids. 

*Pseudolycoriella hygida* larvae build individual cocoons, which is similar to most other sciarid species [102,138] but differs from some *Rhynchosciara* species, which build communal cocoons [102,118]. Descriptions of *Psl. hygida* adult external morphology [25], embryonic development [43], and the pattern of larval eyespot formation at the end of the fourth larval stage [34] have previously been published. The redescription of *Psl. hygida* adults and the description of the immature stages complete the characterization of all developmental stages of this sciarid species and constitute an important resource for future studies on this unconventional model organism. 

### 4.2. Unknown DNA Sequences in Pseudolycoriella: A Challenge in Phylogenetic Systematics 

The COI analysis of 57 sciarid species showed that *Psl. hygida* appearing as sister species of *Psl. nodulosa* + *Psl. bruckii* (Figure 10) and the nearest neighbor species could be *Psl. bruckii* (Winnertz, 1867), with an interspecific distance of 16.64% (Appendix A). Both the large genetic distance and the Palaearctic distribution of *Psl. bruckii*, which is restricted to Europe, indicate the weak points of the ML tree presented in Figure 10. (A) COI sequences alone are generally not suitable for inferring phylogenetically closely related sister species. (B) Our analysis considers only 44 of 161 *Pseudolycoriella* species whose COI barcodes are known (27.3%). In contrast, 72.7% of all described *Pseudolycoriella* species have not yet been genetically analyzed, particularly the species from the tropical and subtropical regions of Central and South America, Africa, and Southeast Asia. (C) Genetic data on ribosomal genes (especially 16S, 18S, 28S) are currently only available for 21.1% of known *Pseudolycoriella* species. Such rDNA sequences have so far only been published for 34 validly described species from Korea (1), Brazil (1), and New Zealand (32) [3,5,50], which makes a well-founded molecular phylogenetic analysis of all described *Pseudolycoriella* impossible at present. For these reasons, the phylogenetic relationships will not be further discussed. On the other hand, a comparison of *Psl. hygida* with morphologically similar *Pseudolycoriella* species discussed in Section 3.5 revealed that it is closely related to *Psl. coecoalata* Mohrig, 2003 from Central America, for which no molecular data are yet available. For the reasons mentioned above, the results presented make an important contribution to the molecular phylogeny of this megadiverse and globally distributed genus. They prove morphologically and genetically that *Psl. hygida* belongs to *Pseudolycoriella* and that the species is more widespread in the Neotropical realm than previously assumed.

### 4.3. Pseudolycoriella hygida as a Model Organism in Cytogenetics

Historically, *Psl. hygida* was the third sciarid species for which polytene chromosome maps were constructed [40] after those of *Bradysia tilicola* (as *Sciara coprophila*) [139] and *Rh. americana* [140]. Together with the maps of *Pss. pubescens* (as *Trichosia*) [141], *Psl. hygida* is the fourth sciarid species for which polytene maps are available in the literature. Chromosome maps are a valuable tool in studies focusing on cytogenetic or molecular traits of species. Their preparation is time-consuming, but the finished map contains a detailed, graphical representation of chromosomes in a given developmental period. The accuracy of polytene maps depends primarily on the chromosome quality in terms of cytological resolution. In this sense, sciarid larvae usually provide polytene chromosomes of very good quality, which are not always present in the salivary gland cells of other dipterans. In some cases, polytene chromosome spreads are below the minimum standard required for image documentation and subsequent map production.

Gene amplification, visualized cytologically in the form of DNA puffs in polytene chromosomes, seems to be common to all the sciarid species that have been studied to date. Although information on the phenomenon is lacking for *Schwenckfeldina* sp., the species included in the present study is also likely to develop DNA puffs, as do other sciarid representatives. In contrast, with regard to gene amplification and RNA expression, *Psl. hygida* is one of the most well-studied organisms.

A comparison of chromosome complements showed that the largest autosome (chromosome A/IV) appeared invariably metacentric/sub-metacentric, while other autosomes and also chromosome X may vary from sub-metacentric to sub-telocentric. The sub-telocentric assumption comes from *Bradysia tilicola* (= *coprophila*), the only species that has been analyzed molecularly in relation to centromere location. Use of specific probes showed that centromere sequences do not reach the very end of chromosome X [49]. Whether other sciarid chromosomes are in fact telocentric remains to be investigated. The conservation of the centromere position observed for the largest autosome is worthy of note, as this mark is shared by distinct genera, i.e., organisms evolutionarily distant albeit belonging to the same dipteran family.

Another conserved characteristic displayed by the sciarid complements involves the NORs, with only one per chromosome complement. Its location in the X chromosome was for a long time considered invariable until the first studies on the subject were carried out on *Psl. hygida*. This species was the first sciarid to have the NOR localized in the chromosome C/II, namely the shortest autosome [36,50]. Years later, an identical NOR location was detected in *Schwenckfeldina* sp. [21]. The evolutionary transfer of the sciarid NOR from the X chromosome to the autosome implies a change in the expression of ribosomal genes, because the two NORs in the autosome pair make dispensable the mechanism of dosage compensation required in sciarid genomes where NORs are on the X chromosome [142].

The comparison of chromosome complements of the species available for analysis supports chromosome evolution unrelated to sciarid phylogeny [5]. It is important to note that given the number of sciarid genera and species in nature, this study is obviously unable to provide a reliable measure of chromosomal features in Sciaridae in relation to either conservation or divergence.

### 4.4. Sciarid Names Used in Cytogenetics and Molecular Studies: Towards an Integrative Approach to Clarifying Scientific Problems

Genetic research on Sciaridae was initiated in the mid-1920s by Charles W. Metz (1889–1975) in the United States, who initially conducted studies on chromosomal behavior and sex determination on *Sciara coprophila*, *S. pauciseta,* and *S. similans* [143,144]. Subsequently, chromosome research was extended to other sciarid species, for which laboratory cultures were established mainly in the United States and Brazil. So far, the following twenty-one sciarid species names have been used to describe study objects in cytogenetics: *americana* Wiedemann, *angelae* Nonato & Pavan, *baschanti* Breuer, *coprophila* Lintner, *elegans* Morgante, *fagilis* Morgante*, hollaenderi* Toledo Filho, *hygida* Sauaia & Alves, *impatiens* Johannsen, *milleri* Pavan & Breuer, *ocellaris* Comstock, *odoriphaga* Yang & Zhang, *tritici* Coquillett, *pauciseta* Felt, *prolifica* Felt, *pubescens* Morgante, *reynoldsi* Metz, *similans* Johannsen, *spatitergum* Hardy, *subtrivialis* Pettey, and *varians* Johannsen. 

Many of the above names were used unchanged in cytogenetic publications over a period of almost 100 years, regardless of the nomenclatural innovations and corrections in the classification of Sciaridae. 

Only the names ***Hybosciara fragilis* Morgante, 1969**, ***Rhynchosciara americana* (Wiedemann, 1821)**, ***Rhynchosciara baschanti* Breuer, 1967**, ***Rhynchosciara hollaenderi* Toledo Filho, 1969**, and ***Rhynchosciara milleri* Pavan & Breuer, 1955** are correct and consistent, because their position in the system of sciarids has not changed over a long period, nor were they shown to be junior synonyms of sciarid taxa described earlier. The type material of ***Bradysia elegans* Morgante, 1969** is still unrevised. Similarly to what happened with *Psl. hygida*, the reexamination of the type series of *B. elegans* may show it to be a species of *Pseudolycoriella*. The remaining taxa belong to thirteen species with the following nomenclatural history (valid species names in bold):

***Bradysia cellarum* Frey, 1948**. Used in cytogenetics as *Bradysia odoriphaga*; *B. odoriphaga* Yang & Zhang, 1985 synonymized by Ye et al. [145]: 183. 

***Bradysia impatiens* (Johannsen, 1912)**. Used in cytogenetics as *Sciara impatiens* or *Bradysia impatiens*; first mentioned as *Bradysia* in Roberts and Lavigne [146]: 25, based on Shaw [147]: 29, 30. 

***Bradysia ocellaris* (Comstock, 1882)**. Used in cytogenetics as *Sciara ocellaris*, *Sciara reynoldsi*, *Sciara tritici*, *Bradysia ocellaris*, or *Bradysia tritici*; *S. ocellaris* combined in the genus *Bradysia* by Tuomikoski [148]: 133; *S. reynoldsi* Metz, 1938 synonymized by Mohrig et al. [7]: 166; *S. tritici* Coquillett, 1895 synonymized by Steffan [149]: 290, with important comments on the type material and nomenclature in Menzel and Mohrig [61]: 155, and Mohrig et al. [7]: 168. 

***Bradysia pallipes* (Fabricius, 1787)**. Used in cytogenetics as *Sciara prolifica*; *S. prolifica* Felt, 1897 synonymized by Mohrig et al. [7]: 168. 

***Bradysia radicum* (Brunetti, 1912)**. Used in cytogenetics as *Bradysia spatitergum*; *B. spatitergum* (Hardy, 1956) synonymized by Mohrig et al. [74]: 410. 

***Bradysia tilicola* (Loew, 1850)**. Used in cytogenetics as *Sciara coprophila* or *Bradysia coprophila*; *S. coprophila* Lintner, 1895 synonymized by Menzel and Mohrig [61]: 148, Menzel and Heller [150]: 231. 

***Bradysia varians* (Johannsen, 1912)**. Used in cytogenetics as *Sciara varians*; first mention as *Bradysia* in Stone and Laffoon [151]: 234. 

***Corynoptera subtrivialis* (Pettey, 1918)**. Used in cytogenetics as *Sciara subtrivialis*; *S. subtrivialis* combined in the genus *Corynoptera* by Steffan [134]: 49. 

***Lycoriella ingenua* (Dufour, 1839)**. Used in cytogenetics as *Sciara pauciseta*; *S. pauciseta* Felt, 1897 synonymized by Steffan [149]: 290, Menzel and Mohrig [61]: 394. 

***Lycoriella sativae* (Johannsen, 1912)**. Used in cytogenetics as *Sciara similans*; *S. similans* Johannsen, 1925 synonymized by Mohrig et al. [7]: 216. 

***Pseudolycoriella hygida* (Sauaia & Alves, 1968)**. Used in cytogenetics as *Bradysia hygida*; *B. hygida* combined in the genus *Pseudolycoriella* by Trinca et al. [26]: 21. 

***Pseudosciara pubescens* (Morgante, 1969)**. Used in cytogenetics as *Trichosia* sp., *Trichosia pubescens,* or *Trichomegalosphys pubescens*; *T. pubescens* combined in the genus *Pseudosciara* by Mohrig and Menzel [8]: 175.

***Rhynchosciara americana* (Wiedemann, 1821)**. Used in cytogenetics as *Rhynchosciara americana* and/or *Rhynchosciara angelae*; *Rh. angelae* Nonato & Pavan, 1951 synonymized by Breuer [152]: 172.

In addition, Brito da Cunha et al. mention a ‘***Bradysia mogiensis***’ in their papers [153,154] that is not included in the catalogue of Neotropical Sciaridae [82]; no valid species description has been found in the literature for this *Bradysia* name. It is said to date back to a publication by the same author [155], in which only a not-further-identified ‘*Bradysia* sp.’ was discussed. Therefore, ‘*Bradysia mogiensis*’ is formally a *nomen nudum* and cannot be used without valid description.

The nomenclatural history of the validated sciarid species does not include a single species that represents the genus *Sciara* Meigen. The cytogenetically analyzed species either belong to other genera or have been given synonymous or invalid names. On the other hand, the inconsistent naming of cytogenetic study objects leads to problems in interdisciplinary research, because after the introduction of genetic analysis methods in phylogenetic systematics, the same reference databases are used by geneticists and taxonomists alike (e.g., BOLD, GenBank). Gene sequences that should be allocated to the same species have been deposited with different species and genus names. Consequently, the inconsistent use of sciarid names has meant that existing datasets are often not recognized or only recognized with difficulty and are therefore either not taken into account in subsequent studies or discussed in the wrong context. For the reasons mentioned above, the correct naming of species used is indispensable and an essential prerequisite for the well-founded comparison of genetic data. This highlights the fact that contemporary scientific problems, such as nomenclatural barriers, can usually only be overcome through the integrated efforts of experts from different research disciplines.

## Figures and Tables

**Figure 1 insects-15-00118-f001:**
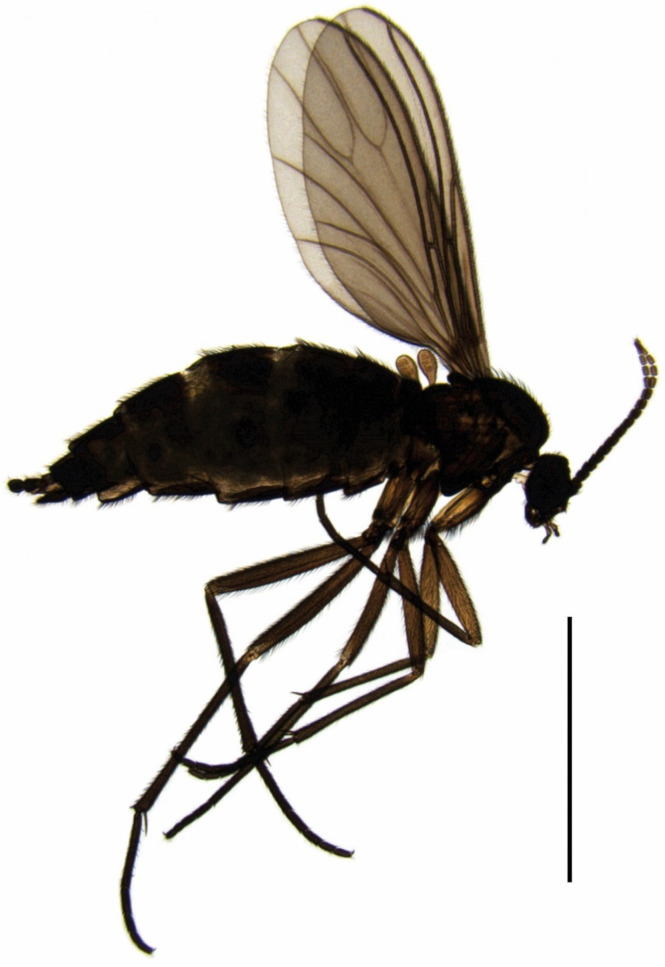
*Pseudolycoriella hygida* (Sauaia & Alves), adult female. Habitus, lateral view. Scale bar: 2 mm.

**Figure 2 insects-15-00118-f002:**
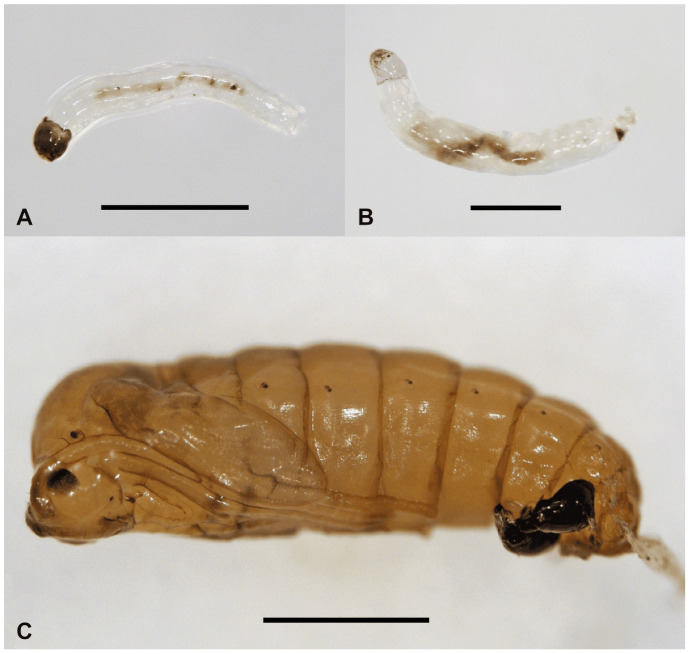
*Pseudolycoriella hygida* (Sauaia & Alves), larva and pupa. (**A**). One-day-old first instar larva. (**B**). Twelve-day-old fourth instar larva; beginning of molt of the head capsule. (**C**). Pupa typically with molted fourth instar sclerotized head capsule still attached to posterior end of abdomen. Scale bars: 500 μm (**A**,**B**); 1 mm (**C**).

**Figure 3 insects-15-00118-f003:**
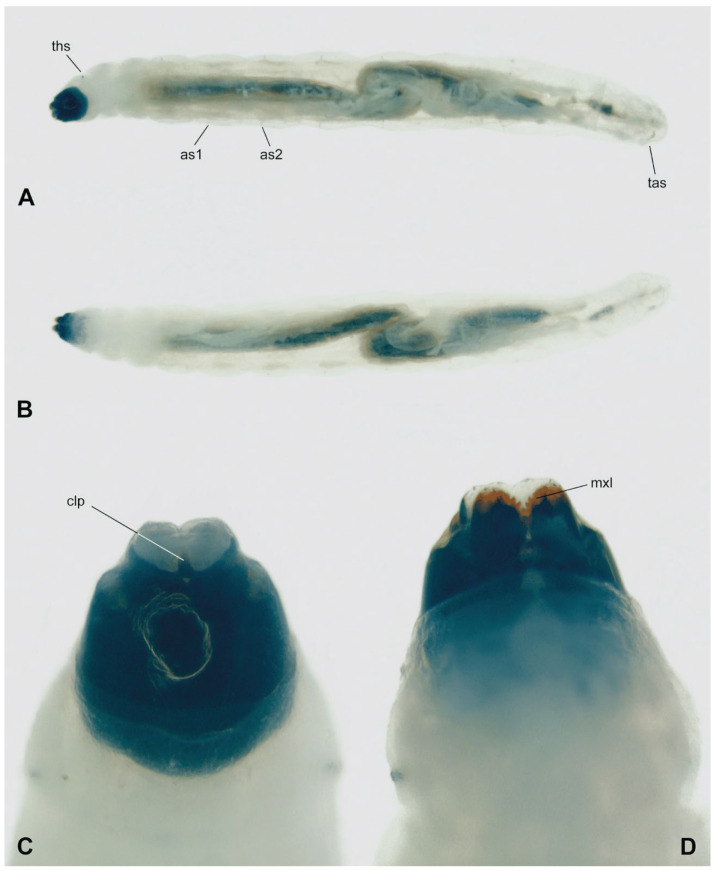
*Pseudolycoriella hygida* (Sauaia & Alves), fourth instar larva. (**A**). Habitus, dorsal view. (**B**). Habitus, ventral view. (**C**). Head capsule, dorsal view. (**D**). Head capsule, ventral view.

**Figure 4 insects-15-00118-f004:**
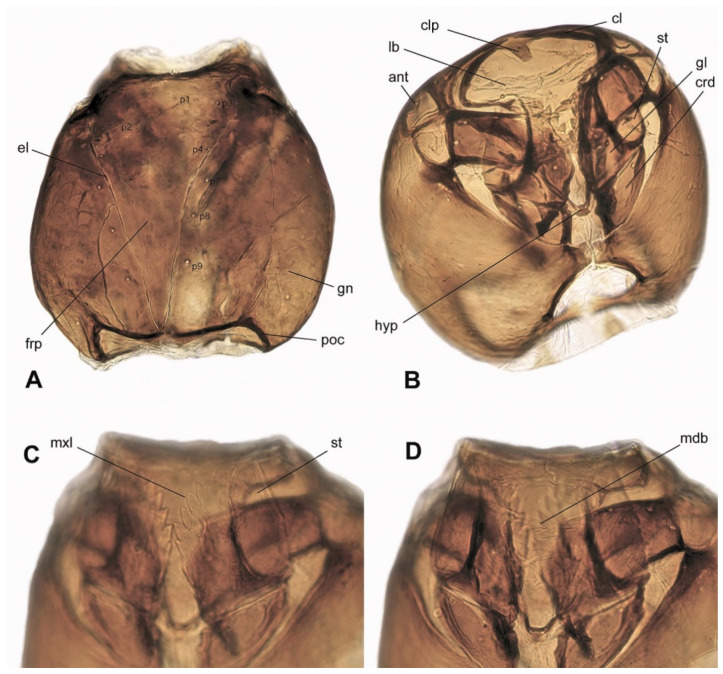
*Pseudolycoriella hygida* (Sauaia & Alves), fourth instar larva. (**A**). Head capsule, dorsoposterior view. (**B**). Head capsule, ventrodistal view. (**C**). Maxilla, ventral view. (**D**). Mandible, ventral view.

**Figure 5 insects-15-00118-f005:**
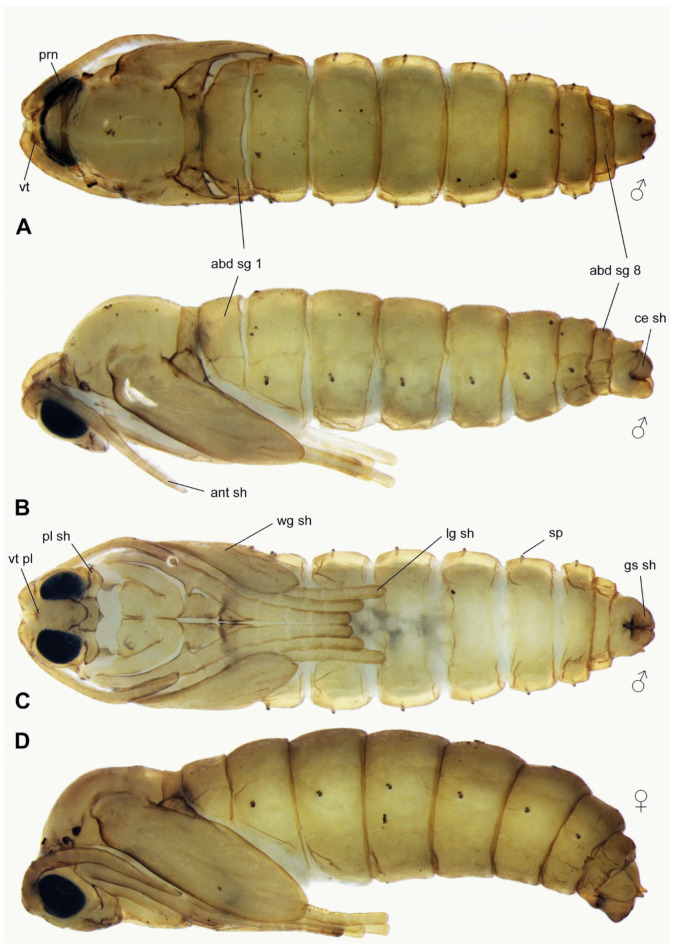
*Pseudolycoriella hygida* (Sauaia & Alves), habitus of pupa. (**A**). Male, dorsal view. (**B**). Male, lateral view. (**C**). Male, ventral view. (**D**). Female, lateral view.

**Figure 6 insects-15-00118-f006:**
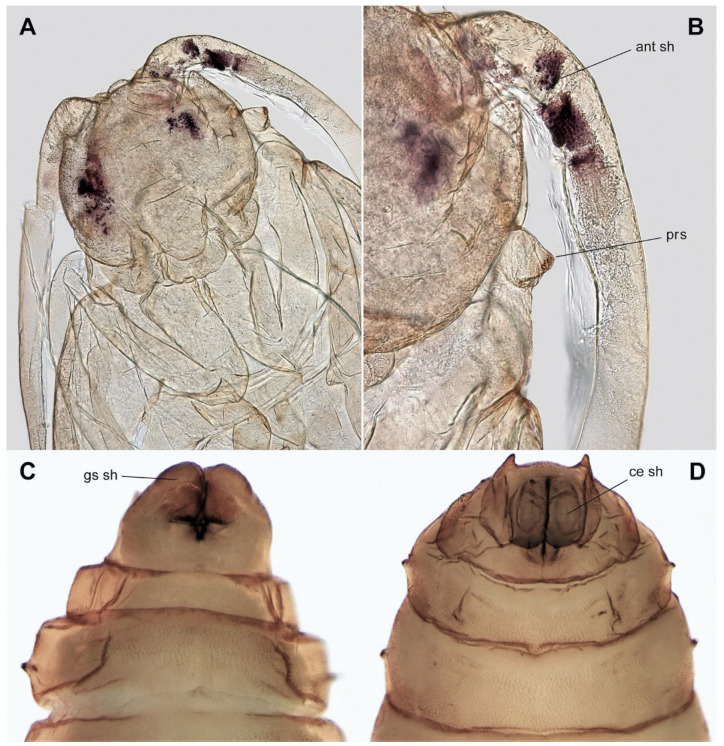
*Pseudolycoriella hygida* (Sauaia & Alves), male pupa. (**A**). Head, ventral view. (**B**). Thorax, ventral view. (**C**). Terminalia, ventral view. (**D**). Terminalia, dorsal view.

**Figure 7 insects-15-00118-f007:**
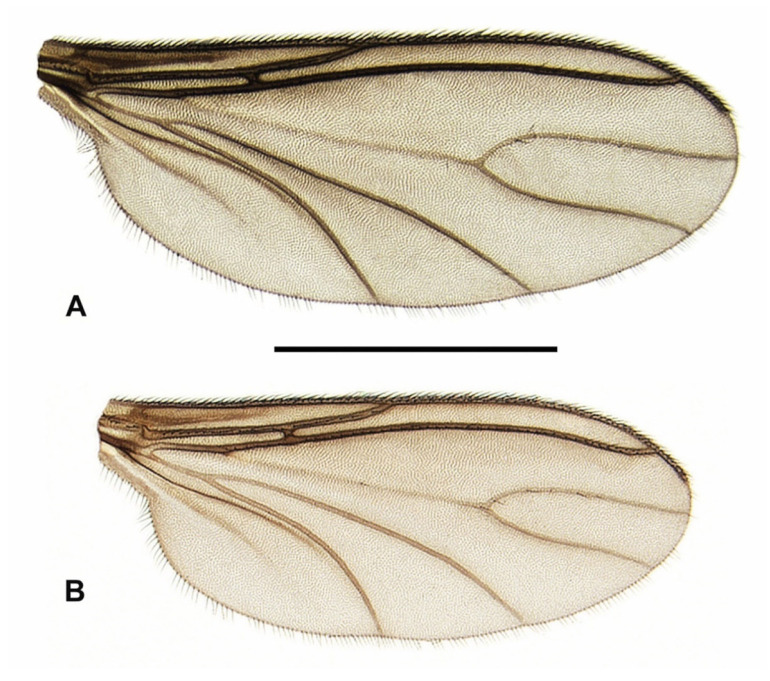
*Pseudolycoriella hygida* (Sauaia & Alves), adults. (**A**). Female wing, dorsal view. (**B**). Male wing, dorsal view. Scale bar: 1 mm.

**Figure 8 insects-15-00118-f008:**
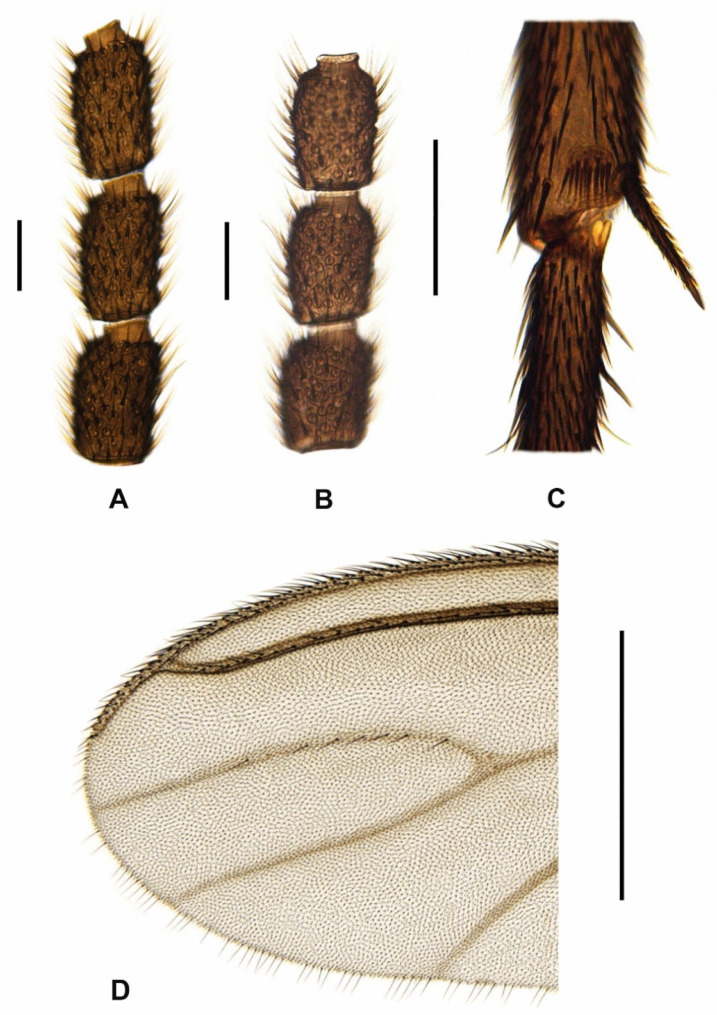
*Pseudolycoriella hygida* (Sauaia & Alves), adults. (**A**). Male flagellomeres 3–5, lateral view. (**B**). Female flagellomeres 3–5, lateral view. (**C**). Apical part of female fore tibia, prolateral view. (**D**). Apex of male wing, dorsal view. Scale bars: 50 μm (**A**,**B**); 100 μm (**C**); 500 μm (**D**).

**Figure 9 insects-15-00118-f009:**
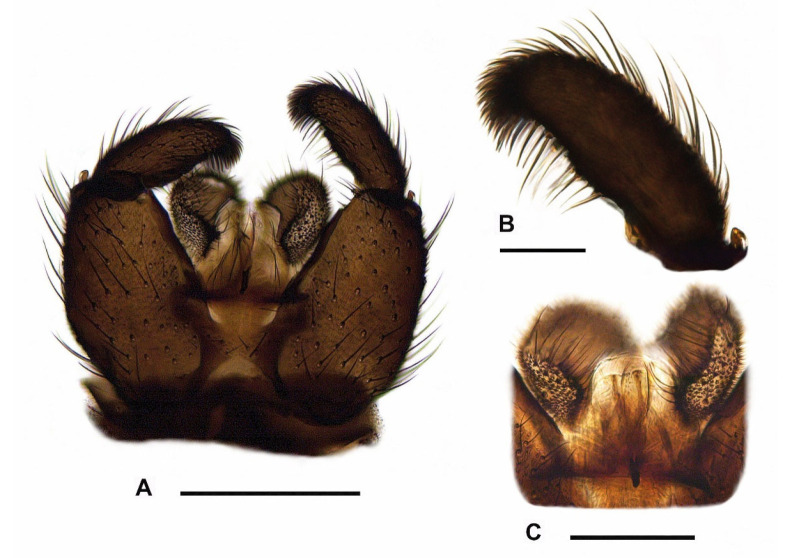
*Pseudolycoriella hygida* (Sauaia & Alves), adult male. (**A**). Hypopygium, ventral view. (**B**). Gonostylus, ventral view; (**C**). Aedeagal complex with tegmen and ejaculatory apodeme, ventral view. Scale bars: 50 μm (**B**); 100 μm (**C**); 200 μm (**A**).

**Figure 10 insects-15-00118-f010:**
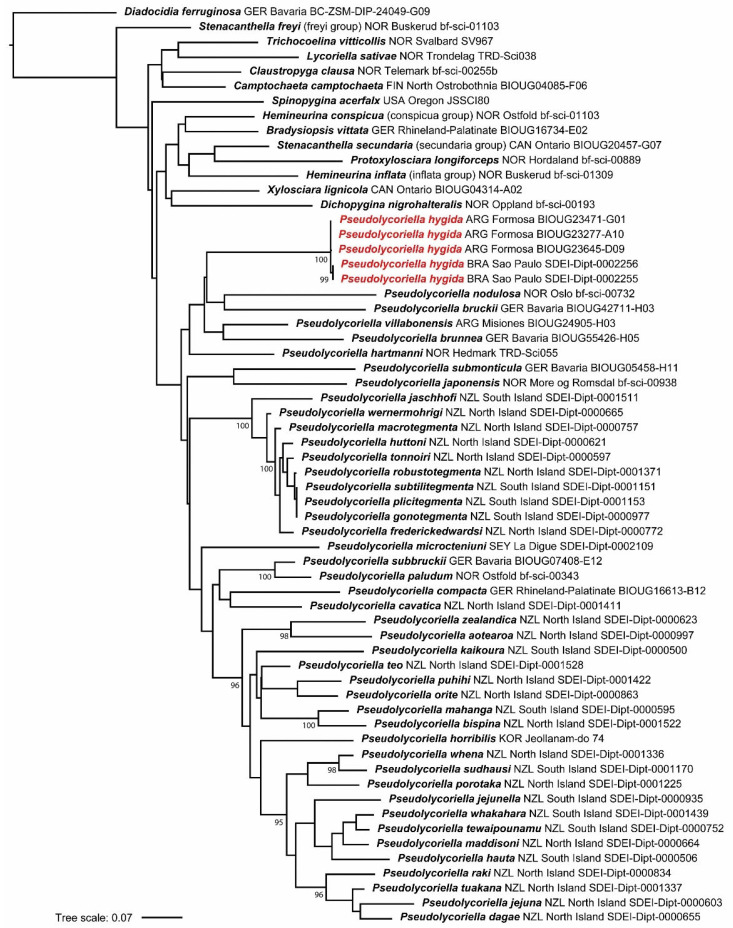
Maximum likelihood tree of *Pseudolycoriella* species and outgroup representatives based on COI barcodes (658 bp). Numbers under branch nodes show UFbootstrap proportions (%). Support values for weakly supported branches (UFboot < 95) are not shown. In red, *Pseudolycoriella hygida*. The scale bar shows the number of estimated substitutions per nucleotide position.

**Figure 11 insects-15-00118-f011:**
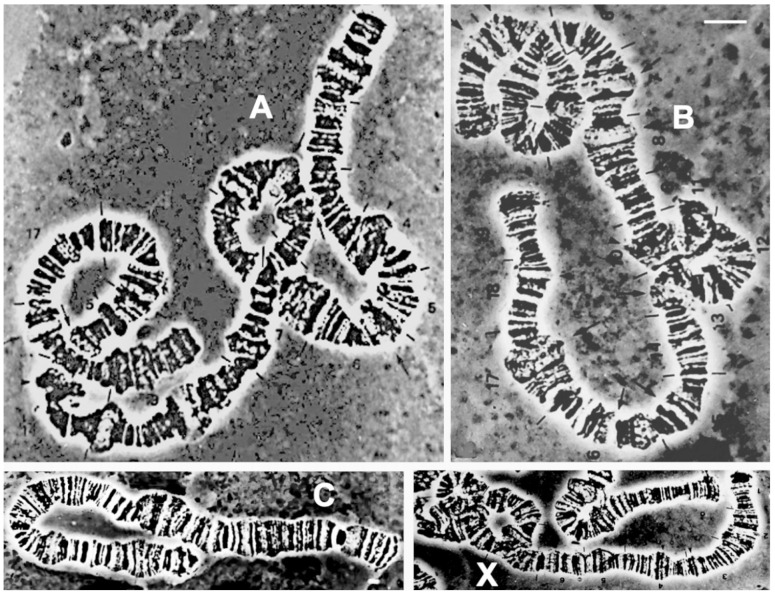
*Pseudolycoriella hygida* (Sauaia & Alves), polytene chromosomes under phase contrast. A, B, C and X indicate chromosomes A, B, C and X, respectively. Numbering along the chromosomes refers to chromosome sections as seen in the polytene maps (Figure 12). Scale bar: 10 μm.

**Figure 12 insects-15-00118-f012:**
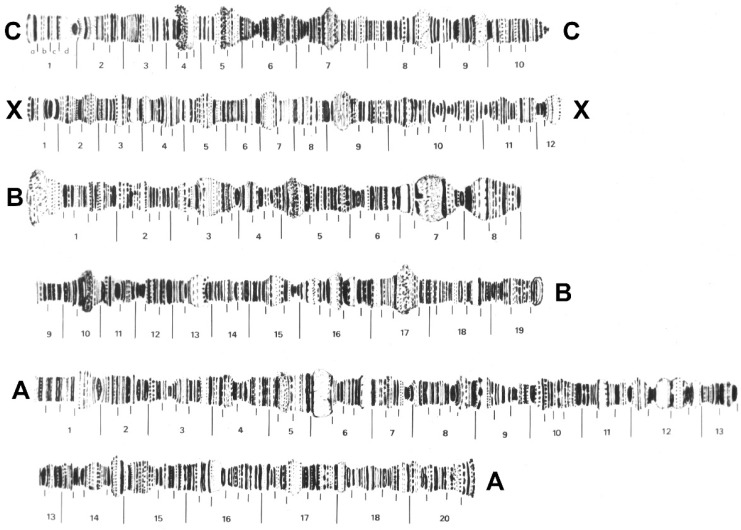
*Pseudolycoriella hygida* (Sauaia & Alves), polytene chromosome maps, slightly modified from Sauaia [40] based on chromosomes from the larval developmental pattern corresponding to the E5 eyespot stage. Autosomes (A–C) and the (X) chromosome are identified. Chromosome sections are numbered and short vertical lines and lowercase letters (example in C1) define chromosome sub-sections.

**Figure 13 insects-15-00118-f013:**
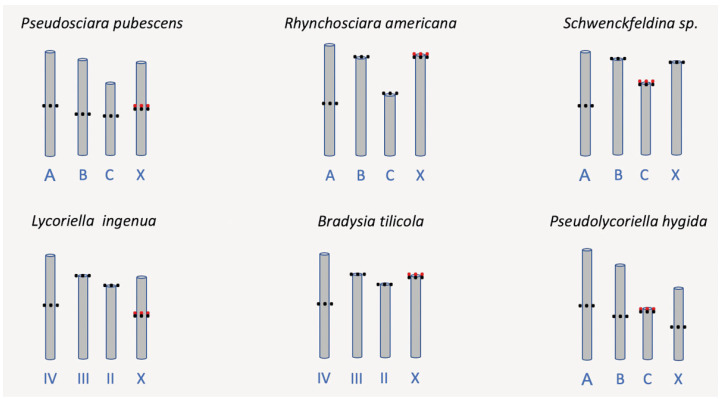
Schematic representation of mitotic/meiotic chromosome complements of six sciarid species showing the location of centromeres (black dots) and nucleolar organizers (NORs) (red dots). Chromosome denominations for *Pss. pubescens*, *Rh. americana*, *Schwenckfeldina sp*., and *Psl. hygida* (A, B, C, X) according to Metz [126], and for *L. ingenua* and *B. tilicola* (IV, III, II, X) according to Crouse and Smith-Stocking [127,129]. Although chromosome length proportions for each sciarid species were maintained, the schemes are not intended to compare chromosome length among different species. References for the preparation of schemes are cited in ‘Material and Methods’, Section 2.2.3.

**Table 1 insects-15-00118-t001:** Collecting sites and reference numbers of the studied *Pseudolycoriella* species and outgroup representatives with COI barcodes from our studies or taken from the BOLD System and GenBank. Taxa are listed alphabetically. Abbreviations used are explained in ‘Material and Methods’, Section 2.5.

Species Name	Collecting Site	Sex	GenBank Accession No.	BOLD BIN	BOLD Sample ID	BOLD Sequence ID	Sample Holder
***Bradysiopsis*** ***vittata*** (Meigen, 1830)	GERMANY: Rhineland-Palatinate	Female	N/A	BOLD:ACC1999	BIOUG16734-E02	GMGME1271-14	ZFMK
***Camptochaeta*** ***camptochaeta*** (Tuomikoski, 1960)	FINLAND: North Ostrobothnia	N/A	N/A	BOLD:ACB7851	BIOUG04085-F06	GMFIC094-12	CBGC
***Claustropyga*** ***clausa*** (Tuomikoski, 1960)	NORWAY: Telemark	Male	N/A	BOLD:ACY8611	bf-sci-00255b	SCINO693-15	BFCO
***Diadocidia*** ***ferruginosa*** (Meigen, 1830)	GERMANY: Bavaria	Male	N/A	BOLD:AAY7756	BC-ZSM-DIP-24049-G09	AMTPD1981-15	ZSMC
***Dichopygina*** ***nigrohalteralis*** (Frey, 1948)	NORWAY: Oppland	Male	KY079365	BOLD:ACM6664	bf-sci-00193	SCINO192-15	BFCO
***Hemineurina*** ***conspicua*** (Winnertz, 1867)	NORWAY: Ostfold	Male	N/A	BOLD:AAP4769	bf-sci-01103	SCINO1356-16	BFCO
***Hemineurina*** ***inflata*** (Winnertz, 1867)	NORWAY: Buskerud	Male	N/A	BOLD:ACJ9929	bf-sci-01309	SCINO1340-16	BFCO
***Lycoriella*** ***sativae*** (Johannsen, 1912)	NORWAY: Trøndelag	Male	N/A	BOLD:ABA1215	TRD-Sci038	SCINO608-15	NTNU
***Protoxylosciara*** ***longiforceps*** (Bukowski & Lengersdorf, 1936)	NORWAY: Hordaland	Male	N/A	BOLD:ACC1367	bf-sci-00889	SCINO1193-16	BFCO
***Pseudolycoriella*** ***aotearoa*** Köhler, 2019	NEW ZEALAND: North Island	Male	MK906359	BOLD:AED8133	SDEI-Dipt-0000997	GBMNB15388-20	SDEI
***Pseudolycoriella*** ***bispina*** Mohrig, 1999	NEW ZEALAND: North Island	Male	MK906401	BOLD:AED6042	SDEI-Dipt-0001522	GBMNB15430-20	SDEI
***Pseudolycoriella*** ***bruckii*** (Winnertz, 1867)	GERMANY: Bavaria	Male	N/A	BOLD:ADW9617	BIOUG42711-H03	GMGMP3875-18	ZSMC
***Pseudolycoriella*** ***brunnea*** (Bukowski & Lengersdorf, 1936)	GERMANY: Bavaria	Male	N/A	BOLD:ACC1590	BIOUG55426-H05	GMGMW520-20	ZSMC
***Pseudolycoriella*** ***cavatica*** (Skuse, 1888)	NEW ZEALAND: North Island	Male	MK906395	BOLD:ABW3602	SDEI-Dipt-0001411	GBMNB15424-20	SDEI
***Pseudolycoriella*** ***compacta*** Heller, 2000	GERMANY: Rhineland-Palatinate	Male	N/A	BOLD:AAN6441	BIOUG16613-B12	GMGMD750-14	ZFMK
***Pseudolycoriella*** ***dagae*** Köhler, 2019	NEW ZEALAND: North Island	Male	MK906331	BOLD:AED4998	SDEI-Dipt-0000655	GBMNB15360-20	SDEI
***Pseudolycoriella*** ***frederickedwardsi*** Köhler, 2016	NEW ZEALAND: North Island	Male	MK906341	BOLD:AED3379	SDEI-Dipt-0000772	GBMNB15370-20	SDEI
***Pseudolycoriella*** ***gonotegmenta*** Köhler, 2019	NEW ZEALAND: South Island	Male	MK906356	BOLD:AED8311	SDEI-Dipt-0000977	GBMNB15385-20	NZAC
***Pseudolycoriella*** ***hartmanni*** (Menzel & Mohrig, 1991)	NORWAY: Hedmark	Male	N/A	BOLD:ACX5047	TRD-Sci055	SCINO625-15	NTNU
***Pseudolycoriella*** ***hauta*** Köhler, 2019	NEW ZEALAND: South Island	Male	MK906322	BOLD:AED1025	SDEI-Dipt-0000506	GBMNB15351-20	SDEI
***Pseudolycoriella*** ***horribilis*** (Edwards, 1931)	SOUTH KOREA: Jeollanam-do	Male	JQ613788	BOLD:ACD4315	74 (JQ613788)	GBMIN12850-13	SNUC
***Pseudolycoriella*** ***huttoni*** Köhler, 2019	NEW ZEALAND: North Island	Male	MK906327	BOLD:AED6374	SDEI-Dipt-0000621	GBMNB15356-20	NZAC
***Pseudolycoriella*** ***hygida*** (Sauaia & Alves, 1968)	BRAZIL: São Paulo	Male	N/A	BOLD:ACW7953	SDEI-Dipt-0002256	SSEY004-23	SDEI
***Pseudolycoriella*** ***hygida*** (Sauaia & Alves, 1968)	BRAZIL: São Paulo	Female	N/A	BOLD:ACW7953	SDEI-Dipt-0002255	SSEY003-23	SDEI
***Pseudolycoriella*** ***hygida*** (Sauaia & Alves, 1968)	ARGENTINA: Formosa	Female	N/A	BOLD:ACW7953	BIOUG23471-G01	GMAFH202-15	MACN
***Pseudolycoriella*** ***hygida*** (Sauaia & Alves, 1968)	ARGENTINA: Formosa	N/A	N/A	BOLD:ACW7953	BIOUG23277-A10	GMAFE075-15	MACN
***Pseudolycoriella*** ***hygida*** (Sauaia & Alves, 1968)	ARGENTINA: Formosa	N/A	N/A	BOLD:ACW7953	BIOUG23645-D09	GMAFL125-15	MACN
***Pseudolycoriella*** ***japonensis*** (Mohrig & Menzel, 1992)	NORWAY: Møre og Romsdal	Male	N/A	BOLD:ACZ8590	bf-sci-00938	SCINO1241-16	BFCO
***Pseudolycoriella*** ***jaschhofi*** Köhler, 2019	NEW ZEALAND: South Island	Male	MK906409	BOLD:AED7085	SDEI-Dipt-0001511	GBMNB15438-20	SDEI
***Pseudolycoriella*** ***jejuna*** (Edwards, 1927)	NEW ZEALAND: North Island	Male	MK906326	BOLD:AED3604	SDEI-Dipt-0000603	GBMNB15355-20	SDEI
***Pseudolycoriella*** ***jejunella*** Köhler, 2019	NEW ZEALAND: South Island	Male	MK906351	BOLD:AED1935	SDEI-Dipt-0000935	GBMNB15380-20	SDEI
***Pseudolycoriella*** ***kaikoura*** Köhler, 2019	NEW ZEALAND: South Island	Male	MK906321	BOLD:AED6542	SDEI-Dipt-0000500	GBMNB15350-20	NZAC
***Pseudolycoriella*** ***macrotegmenta*** Mohrig, 1999	NEW ZEALAND: North Island	Male	MK906340	BOLD:AED4092	SDEI-Dipt-0000757	GBMNB15369-20	SDEI
***Pseudolycoriella*** ***maddisoni*** Köhler, 2019	NEW ZEALAND: North Island	Male	MK906332	BOLD:AED4438	SDEI-Dipt-0000664	GBMNB15361-20	SDEI
***Pseudolycoriella*** ***mahanga*** Köhler, 2019	NEW ZEALAND: South Island	Male	MK906324	BOLD:AED1463	SDEI-Dipt-0000595	GBMNB15353-20	SDEI
***Pseudolycoriella*** ***microcteniuni*** (Yang & Zhang, 1987)	SEYCHELLES: La Digue	Male	N/A	BOLD:AFG5550	SDEI-Dipt-0002109	SSEY002-23	SDEI
***Pseudolycoriella*** ***nodulosa*** (Mohrig &Krivosheina, 1985)	NORWAY: Oslo	Male	N/A	BOLD:ACY8397	bf-sci-00732	SCINO772-15	BFCO
***Pseudolycoriella*** ***orite*** Köhler, 2019	NEW ZEALAND: North Island	Male	MK906347	BOLD:AED1715	SDEI-Dipt-0000863	GBMNB15376-20	NZAC
***Pseudolycoriella*** ***paludum*** (Frey, 1948)	NORWAY: Østfold	Male	N/A	BOLD:ACP4204	bf-sci-00343	SCINO341-15	BFCO
***Pseudolycoriella*** ***plicitegmenta*** Köhler, 2019	NEW ZEALAND: South Island	Male	MK906366	BOLD:AED8311	SDEI-Dipt-0001153	GBMNB15395-20	SDEI
***Pseudolycoriella*** ***porotaka*** Köhler, 2019	NEW ZEALAND: North Island	Male	MK906375	BOLD:AED5151	SDEI-Dipt-0001225	GBMNB15404-20	NZAC
***Pseudolycoriella*** ***puhihi*** Köhler, 2019	NEW ZEALAND: North Island	Male	MK906396	BOLD:AED1266	SDEI-Dipt-0001422	GBMNB15425-20	SDEI
***Pseudolycoriella*** ***raki*** Köhler, 2019	NEW ZEALAND: North Island	Male	MK906345	BOLD:AED4711	SDEI-Dipt-0000834	GBMNB15374-20	NZAC
***Pseudolycoriella*** ***robustotegmenta*** Köhler, 2019	NEW ZEALAND: North Island	Male	MK906393	BOLD:AED8311	SDEI-Dipt-0001371	GBMNB15422-20	SDEI
***Pseudolycoriella*** ***subbruckii*** (Mohrig &Hövemeyer, 1992)	GERMANY: Bavaria	Male	N/A	BOLD:ACG6269	BIOUG07408-E12	GMGRF345-13	ZSMC
***Pseudolycoriella*** ***submonticula*** (Mohrig & Mamaev, 1990)	GERMANY: Bavaria	Female	N/A	BOLD:ACF9727	BIOUG05458-H11	GMGRC1885-13	ZSMC
***Pseudolycoriella*** ***subtilitegmenta*** Köhler, 2019	NEW ZEALAND: South Island	Male	MK906364	BOLD:AED8311	SDEI-Dipt-0001151	GBMNB15393-20	NZAC
***Pseudolycoriella*** ***sudhausi*** Köhler, 2019	NEW ZEALAND: South Island	Male	MK906369	BOLD:AED1862	SDEI-Dipt-0001170	GBMNB15398-20	SDEI
***Pseudolycoriella*** ***teo*** Köhler, 2019	NEW ZEALAND: North Island	Male	MK906402	N/A	SDEI-Dipt-0001528	GBMNB15431-20	NZAC
***Pseudolycoriella*** ***tewaipounamu*** Köhler, 2019	NEW ZEALAND: South Island	Male	MK906338	BOLD:AED4891	SDEI-Dipt-0000752	GBMNB15367-20	SDEI
***Pseudolycoriella*** ***tonnoiri*** Köhler, 2016	NEW ZEALAND: North Island	Male	MK906325	BOLD:ADM7629	SDEI-Dipt-0000597	GBMNB15354-20	SDEI
***Pseudolycoriella*** ***tuakana*** Köhler, 2019	NEW ZEALAND: North Island	Male	MK906389	BOLD:AED4777	SDEI-Dipt-0001337	GBMNB15418-20	SDEI
***Pseudolycoriella*** ***villabonensis*** Heller, 2012	ARGENTINA: Misiones	N/A	OM597266	BOLD:ACO0105	BIOUG24905-H03	GMAGO963-15	MACN
***Pseudolycoriella*** ***wernermohrigi*** Köhler, 2019	NEW ZEALAND: North Island	Male	MK906333	BOLD:AED4093	SDEI-Dipt-0000665	GBMNB15362-20	SDEI
***Pseudolycoriella*** ***whakahara*** Köhler, 2019	NEW ZEALAND: South Island	Male	MK906408	BOLD:AED2160	SDEI-Dipt-0001439	GBMNB15437-20	NZAC
***Pseudolycoriella*** ***whena*** Köhler, 2019	NEW ZEALAND: North Island	Male	MK906388	BOLD:AED1474	SDEI-Dipt-0001336	GBMNB15417-20	NZAC
***Pseudolycoriella*** ***zealandica*** (Edwards, 1927)	NEW ZEALAND: North Island	Male	MK906328	BOLD:ADN2926	SDEI-Dipt-0000623	GBMNB15357-20	SDEI
***Spinopygina acerfalx*** Vilkamaa, Burdíková & Ševčík, 2023	USA: Oregon	Male	OQ024760	N/A	JSSCI80	N/A	UOLC
***Stenacanthella*** ***freyi*** (Tuomikoski, 1960)	NORWAY: Buskerud	Male	N/A	BOLD:ACU4800	bf-sci-00397	SCINO395-15	BFCO
***Stenacanthella*** ***secundaria*** (Mohrig & Menzel, 1990)	CANADA: Ontario	Male	MF841640	BOLD:AAU6614	BIOUG20457-G07	CNTIA197-15	CBGC
***Trichocoelina*** ***vitticollis*** (Holmgren, 1883)	NORWAY: Svalbard	Male	MN135691	BOLD:ABA5288	SV967	SVDIP306-13	NHMO
***Xylosciara*** ***lignicola*** (Winnertz, 1867)	CANADA: Ontario	Female	KR432701	BOLD:AAU6615	BIOUG04314-A02	CNBPD380-12	CBGC

## Data Availability

Data are available in tables and figures, including Appendix A. The COI barcodes of *Pseudolycoriella hygida* (IDs SDEI-Dipt-0002255, SDEI-Dipt-0002256) and *Pseudolycoriella microcteniuni* (ID SDEI-Dipt-0002109) generated for this study have been uploaded on BOLD and are publicly available.

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
