# Peer review of "Pseudolycoriella hygida (Sauaia and Alves)—An Overview of a Model Organism in Genetics, with New Aspects in Morphology and Systematics"

_insects, 2024, doi:10.3390/insects15020118_

Round 1

Reviewer 1 Report

Comments and Suggestions for Authors

The manuscript provides an overview of the current knowledge in taxonomy, systematics, and physiology of the species Pseudolycoriella hygida. In the study the authors use several methods, including rearing, microscopy and DNA sequencing. The introduction is thorough but lacks focus. Some parts are repetitive and can be shortened. Many parts of the results and discussion seems out of place and should be moved to the introduction. Overall, it feels unclear whether the authors intend to present a literature review or the results of an experiment. The results from sequencing and morphological analysis are presented in detail, but not discussed and put into relation to previous research. I have given a number of suggestions within the manuscript where I also give further and more detailed comments. In general, the manuscript is interesting and contain parts of high scientific value with a sound basis, but lacks structure and clear objectives and presentation of results.

Reviewer 2 Report

Comments and Suggestions for Authors

Dear authors,

it was my pleasure to read the manuscript. The amount of information you presented in the paper is impressive. Keep on with a good work. One  suggestion, I would encourage you to analyze more genes in the future (for example nuclear genes) in order to get more insight into phylogeny within the genus, of the genus and even higher taxonomic ranks.  It would complement the shortcomings in the field of study of family Sciaridae. 

I attach PDF with my comments.

Introduction

Line 54- I suppose that the number of the reference(s) is sufficient to be in the text, and not the author(s) name(s).

Results

Line 411- Perhaps, a scale bar should be added to the photo, if possible.

Line 434- Please add larva length (height, width)

Line 461- Please add pupa length (height, width)

Line 720- Perhaps this table should be shown as Supplementary material, if possible, it is too long.

Line 696-710- this part should be within the Discussion section.

Discussion

Line 943- since you presented the ML tree based on COI (in the Results), maybe it would be useful to add a small paragraph on this. You can use the part of the paragraph in the Results section (Line 696-710).

Round 2

Reviewer 1 Report

Comments and Suggestions for Authors

The manuscript provides an overview of the current knowledge in taxonomy, systematics, and physiology of the species Pseudolycoriella hygida. In the study the authors use several methods, including rearing, microscopy and DNA sequencing.

This is the second review after I recommended a major revision and left a number of recommendations and comments. The main issues of the first version were that it was unfocused with a lack of structure and parts of the text in unsuitable places in the manuscript.

The authors have done a throrough work with adressing all my suggestions, and have given fully acceptable reasons in the few cases they have disregarded them.

The manuscript has a much more increased focus and the introduction as well as the discussion is better structured. The results have been cleaned up to only present the results of the present study rather than review of previous litterature.

I would recommend the manuscript for publication in it's current state.